# EFFICIENT SPECTRAL GRAPH DIFFUSION BASED ON SYMMETRIC NORMALIZED LAPLACIAN

## ABSTRACT

Graph generative modeling has seen rapid progress, yet existing approaches often trade off between fidelity, scalability, and stability. Continuous and discrete diffusion models capture complementary aspects but remain hampered by either structural distortion or heavy computational costs. We introduce Efficient Spectral Graph Diffusion (ESGD), a lightweight framework that performs diffusion in the compressed eigenvalue space of the Symmetric Normalized Laplacian (SNL). This spectral compression guarantees bounded eigenvalues, provable stability, and faster convergence while eliminating hub-node dominance. A novel degree-matrix recovery algorithm enables exact graph reconstruction from the spectral representation. ESGD achieves state-of-the-art generation quality with one of the smallest parameter counts, converging up to 100× faster in training and requiring 6–10× fewer sampling steps with up to 2000× less computational cost. Our findings suggest that progress in graph generation may come less from heavier engineering, and more from principled reformulations that unlock both efficiency and fidelity.

## 1 INTRODUCTION

Graph distribution learning and generation have become central research topics with broad applications in drug discovery, materials science, and network analysis. The goal is to capture the underlying distribution of graphs and model their intrinsic structural properties, including the interplay between nodes, edges, and features. Early generative models such as variational autoencoders (GraphVAE Simonovsky & Komodakis (2018)) and generative adversarial networks De Cao & Kipf (2018); Miyato et al. (2018) demonstrated feasibility, but VAEs struggle with posterior estimation on large graphs, while GANs are prone to mode collapse Jo et al. (2022). These limitations highlight the need for more scalable and robust paradigms.

Diffusion-based approaches have recently shown remarkable promise. Early models operate directly on adjacency matrices or their eigenspaces, applying Gaussian perturbations to both node features and graph structure Niu et al. (2020b); Jo et al. (2022). To preserve sparsity and improve efficiency, discrete diffusion models such as DiGress Vignac et al. (2023b) and DeFoG Qin et al. (2025) introduce edit-based noise processes. In addition to discrete models, Laplacian Martinkus et al. (2022b); Bergmeister et al. (2024a) and spectral Luo et al. (2024); Minello et al. (2025) methods which explore diffusion over eigenvalues and eigenvectors, capturing global structural properties but often suffering from eigenvalue imbalance or added model complexity.

In this paper, we propose ESGD, a framework that addresses three fundamental challenges in spectral graph generation. First, we compress eigenvalues of the SNL into the bounded interval [-1, 1], which eliminates the dependence on maximum node degree and provides uniform spectral radius bounds regardless of graph topology. This compression yields theoretical guarantees: bounded signal-to-noise ratios, and improved Lipschitz constants for score functions. Second, we develop a degree-matrix recovery algorithm that uniquely reconstructs the degree matrix from the compressed spectral representation, closing the reconstruction gap that limited prior spectral methods. Third, we demonstrate scalability to citation networks through ego-subgraph decomposition, achieving strong performance on larger graphs and show potentials on graphs with thousands of nodes while maintaining computational tractability. Empirically, ESGD achieves competitive or superior generation quality compared to both spectral and discrete diffusion baselines, while requiring orders of mag-

nitude fewer computational resources. Figure 1a intuitively demonstrates the superiority of ESGD over other baselines in terms of generation efficiency and quality.

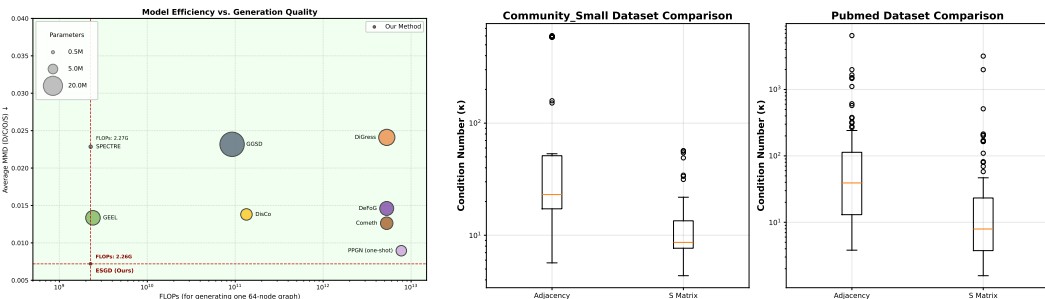

(a) Model comparison on Planar dataset      (b) Condition Number Distributions: Adjacency vs S Matrix

Figure 1: (a) Comparison of generation efficiency and quality between ESGD and baseline models from recent three years on the Planar dataset. (b) ESGD transforms the adjacency matrix into an SNL matrix (S matrix), suppresses extreme values and yielded a more well-behaved data distribution therefore improving spectral properties.

## 2 RELATED WORK

**Early graph generative models.** Early approaches relied on VAEs and GANs. GraphVAE Simonovsky & Komodakis (2018) and MolGAN De Cao & Kipf (2018); Miyato et al. (2018) showed that deep generative learning on graphs is possible, but inherited the weaknesses of their backbones: VAEs suffer from inaccurate posterior approximation on large graphs, while GANs are prone to instability and mode collapse Jo et al. (2022). These limitations motivated the search for more stable generative paradigms.

**Diffusion-based generative modeling.** Diffusion models—including DDPM Ho et al. (2020), DDIM Song et al. (2021a), score-based diffusion Song et al. (2021b), stable diffusion Rombach et al. (2022), and flow-based variants Lipman et al. (2023); Liu et al. (2022)—have since emerged as a powerful family of generative methods, overcoming many of the weaknesses of VAEs and GANs in high-dimensional domains. Their success in images and molecules has spurred growing interest in graphs, where two main directions have been explored:

*Continuous diffusion.* These models Niu et al. (2020b); Jo et al. (2022) apply Gaussian perturbations to adjacency matrices and node features. While effective, the injected noise often produces dense graphs, degrading sparsity and structural fidelity.

*Discrete diffusion.* In contrast, models such as DiGress Vignac et al. (2023b), EDGE Chen et al. (2023a), local-PPGN Bergmeister et al. (2024b) GEEL Jang et al. (2024) DeFoG Qin et al. (2025) define edit-based noise processes on nodes and edges, preserving sparsity and graph structure. However, they require long training schedules and slow sampling, which limits scalability.

**Spectral and Laplacian approaches.** A complementary line of work leverages graph spectra. SPECTRE Martinkus et al. (2022b) models dominant Laplacian eigencomponents to capture global structure but introduces significant architectural complexity. GSDM Luo et al. (2024) improves efficiency via low-rank spectral diffusion, yet remains sensitive to eigenvalue scaling. GGSD Minello et al. (2025) increases complexity by sampling eigencomponents. These methods illustrate the potential of spectrum, but also reveal persistent limitations in stability, reconstruction, and scalability.

**Our approach.** ESGD addresses these limitations via eigenvalue compression of the symmetric normalized Laplacian, bounding all eigenvalues to [-1,1] and eliminating degree-dependent scaling. The SNL's symmetry naturally aligns with diffusion models' zero-mean Gaussian priors, simplifying optimization. By diffusing only eigenvalues while fixing eigenvectors, ESGD achieves competitive quality with orders of magnitude lower computational cost.

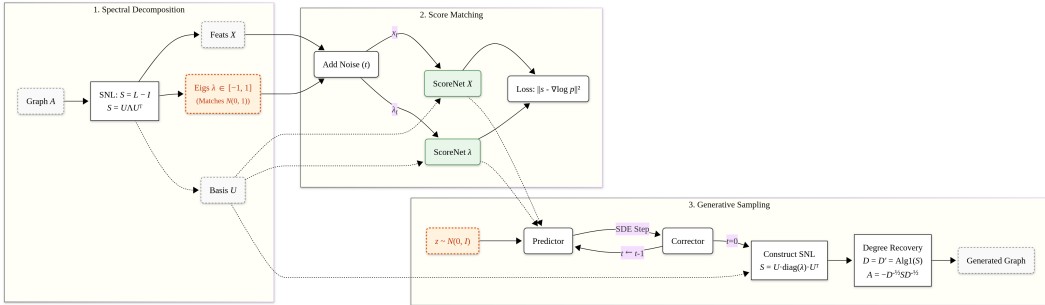

Figure 2: ESGD pipeline. The spectral decomposition phase computes the SNL matrix for all graphs and extracts eigenvectors $U$, eigenvalues $\Lambda$ ($\text{diag}(\lambda)$), and node features $X$. The score matching phase trains two networks to denoise $X$ and $\Lambda$. The generation phase reconstructs the adjacency matrix: generated eigenvalues $\hat{\Lambda}$ combine with fixed basis $U$ to form the predicted SNL matrix, from which Algorithm 1 recovers the degree matrix and adjacency $\hat{A}$. The final graph pairs generated node features $\hat{X}$ with reconstructed $\hat{A}$.

# 3 PRELIMINARIES

## 3.1 SPECTRAL PROPERTIES AND CONDITION NUMBERS

For a matrix $M$ with eigenvalues $\lambda_1, \ldots, \lambda_n$, the condition number is defined as $\kappa(M) = \frac{|\lambda_{\max}|}{|\lambda_{\min}|}$, measuring the ratio between the largest and smallest eigenvalue magnitudes. A high condition number indicates that the matrix is ill-conditioned, meaning small perturbations in the input can lead to large changes in the output. In diffusion models, the condition number of the data covariance matrix directly affects optimization stability and convergence speed. In continuous diffusion models, the eigenvalues typically scale as the maximum node degree $\Delta_{\max}$. For scale-free graphs, $\Delta_{\max}$ can be very large, which induces severe information imbalance in the diffusion model: high-degree hub nodes dominate the learning signal, while low-degree nodes receive insufficient gradient updates.

Moreover, in a dataset, the unbalance distribution in condition number means these special graphs are hard to learn, see Figure 1b. This issue affects not only continuous diffusion models built on the adjacency spectrum (GGSD, GSDM), but also discrete diffusion models (Digress, DisCo, DeFoG, Cometh). In discrete models, all graphs in the dataset are considered and the model is trained to predict node labels one by one. Consequently, the model has to accommodate the influence of outlier graphs in the data, which gives rise to two undesired effects: (i) degraded predictive performance, and (ii) increased training cost in terms of both optimization steps and model size, see section 6.4.

## 3.2 SCORE-BASED GENERATIVE MODELS

Diffusion-based generative modeling has emerged as a powerful paradigm for high-dimensional data generation. In score-based generative models Song et al. (2021c), the key idea is to learn the score function $\nabla_{\boldsymbol{z}} \log p_t(\boldsymbol{z})$, the gradient of the log-density of a perturbed data distribution at time $t$. This simulates the reverse-time stochastic differential equation (SDE) to transform noise into data samples.

Formally, the forward noising process is defined by an SDE

$$d\boldsymbol{z}_t = f(t, \boldsymbol{z}_t)dt + g(t)d\boldsymbol{w}_t, \quad t \in [0, 1], \tag{1}$$

where $f$ and $g$ denote drift and diffusion coefficients, $\boldsymbol{w}_t$ is a standard Wiener process. As $t \to 1$, $\boldsymbol{z}_t$ converges to a simple prior distribution (e.g., Gaussian). The reverse-time SDE takes the form

$$d\boldsymbol{z}_t = \Big(f(t, \boldsymbol{z}_t) - g(t)^2 \nabla_{\boldsymbol{z}} \log p_t(\boldsymbol{z}_t)\Big)dt + g(t)d\bar{\boldsymbol{w}}_t, \tag{2}$$

where $\bar{\boldsymbol{w}}_t$ is a reverse-time Wiener process. In practice, the score function is unknown and must be approximated by a neural network $s_{\boldsymbol{\theta}}(\boldsymbol{z}_t, t)$.

### 3.3 Graph Convolutional Networks (GCN)

Graph Convolutional Networks (GCNs) Kipf & Welling (2017) are a fundamental building block for learning on graphs. Given an undirected graph with adjacency matrix $\boldsymbol{A}$ and degree matrix $\boldsymbol{D}$, GCN defines a layer-wise propagation rule that aggregates information from neighbors:

$$\boldsymbol{H}^{(\ell+1)} = \sigma\left(\hat{\boldsymbol{A}}\,\boldsymbol{H}^{(\ell)}\,\boldsymbol{W}^{(\ell)}\right), \hat{\boldsymbol{A}} = \tilde{\boldsymbol{D}}^{-1/2}\,\tilde{\boldsymbol{A}}\,\tilde{\boldsymbol{D}}^{-1/2}, \quad \tilde{\boldsymbol{A}} = \boldsymbol{A} + \boldsymbol{I}. \tag{3}$$

where $\boldsymbol{H}^{(\ell)}$ is the hidden representation at layer $\ell$, $\boldsymbol{W}^{(\ell)}$ is a trainable weight matrix, $\sigma$ is a non-linear activation

## 4 Methodology

We study the graph generation problem where each graph $G = (\boldsymbol{X}, \boldsymbol{A})$ consists of a node-feature matrix $\boldsymbol{X} \in \mathbb{R}^{n \times d}$ and an adjacency matrix $\boldsymbol{A}$. We define the SNL operator

$$\boldsymbol{U}\boldsymbol{\Lambda}\boldsymbol{U}^\top = \boldsymbol{S} = \boldsymbol{L} - \boldsymbol{I} = -\boldsymbol{D}^{-\frac{1}{2}}\boldsymbol{A}\boldsymbol{D}^{-\frac{1}{2}},$$

Let $\boldsymbol{S} = \boldsymbol{U}\boldsymbol{\Lambda}\boldsymbol{U}^\top$ be its eigen-decomposition. ESGD performs diffusion in a fixed eigen-space, we keep $\boldsymbol{U}$ fixed and only diffuse the eigenvalues $\boldsymbol{\Lambda}$ together with node features $\boldsymbol{X}$. The forward process is given by two coupled SDEs:

$$d\boldsymbol{X}_t = f_{\boldsymbol{X}}(\boldsymbol{X}_t, t)dt + g_{\boldsymbol{X}}(t)d\boldsymbol{W}_t^{\boldsymbol{X}}, \quad d\boldsymbol{\Lambda}_t = f_{\boldsymbol{\Lambda}}(\boldsymbol{\Lambda}_t, t)dt + g_{\boldsymbol{\Lambda}}(t)d\boldsymbol{W}_t^{\boldsymbol{\Lambda}},$$

with independent Wiener processes for $\boldsymbol{X}$ and $\boldsymbol{\Lambda}$. The reverse SDEs follow standard score-based formulations using score networks $s_{\boldsymbol{\theta}}$ and $s_{\boldsymbol{\phi}}$.

**Score Networks** We use GCN-based architectures for both $s_{\boldsymbol{\theta}}$ (node features) and $s_{\boldsymbol{\phi}}$ (eigenvalues). To avoid double normalization and self loop, the message-passing operator is replaced by $-\boldsymbol{S}$:

$$\boldsymbol{H}^{(\ell+1)} = \sigma(-\boldsymbol{S}\boldsymbol{H}^{(\ell)}\boldsymbol{W}^{(\ell)}).$$

**Objectives** We minimize denoising score matching losses:

$$\hat{E}(\boldsymbol{\theta}) = \mathbb{E}\|s_{\boldsymbol{\theta}}(\boldsymbol{X}_t, \boldsymbol{\Lambda}_t, \boldsymbol{U}) - \nabla_{\boldsymbol{X}_t} \log p_t(\boldsymbol{X}_t|\boldsymbol{X}_0)\|^2,$$
$$\hat{E}(\boldsymbol{\phi}) = \mathbb{E}\|s_{\boldsymbol{\phi}}(\boldsymbol{X}_t, \boldsymbol{\Lambda}_t, \boldsymbol{U}) - \nabla_{\boldsymbol{\Lambda}_t} \log p_t(\boldsymbol{\Lambda}_t|\boldsymbol{\Lambda}_0)\|^2.$$

**Sampling** After training, we reverse the diffusion to obtain $(\hat{\boldsymbol{X}}_0, \hat{\boldsymbol{\Lambda}}_0)$. The adjacency matrix is reconstructed using the recovered eigenvalues $\hat{\boldsymbol{\Lambda}}$:

$$\hat{\boldsymbol{A}} = -\boldsymbol{D}^{1/2}\hat{\boldsymbol{S}}\boldsymbol{D}^{1/2}, \quad \hat{\boldsymbol{S}} = \boldsymbol{U}\hat{\boldsymbol{\Lambda}}\boldsymbol{U}^\top,$$

where the degree matrix $\boldsymbol{D}$ is uniquely recoverable from $\hat{\boldsymbol{S}}$ by a provable algorithm 1. This guarantees exact graph reconstruction up to numerical thresholds. The full details of the ESGD model architecture can be found in B.2 and supplementary code project files. The ESGD pipeline is illustrated in Figure 2.

## 5 Theoretical Properties

As we discussed in subsection 3.1, the spectral properties from graph theory influences the models at the first beginning. Beyond eigenvalue normalization, the symmetry of the operator is particularly well aligned with diffusion models. In standard diffusion setups, the prior distribution is a standard Gaussian with zero mean and unit variance. This symmetry reduces the burden on the model, since it does not need to learn the global mean of the data distribution and can instead focus on higher-order structure. As a result, optimization becomes easier and computation can be accelerated while improving the performance. More discussion can be seen in 6 with different aspects: Performance in 6.1 6.3, Efficiency in 6.4, Generalization in 6.2.

We establish three key results: eigenvalue boundedness in Theorem 5.1, information-theoretic advantages in Theorem 5.3, and optimization stability in Theorem 5.4, 5.5. See Appendix A for detailed proofs.

**Theorem 5.1** (Spectral boundedness). *For an undirected graph with adjacency matrix $\boldsymbol{A}$ and symmetric normalized Laplacian $\boldsymbol{S} = \boldsymbol{L} - \boldsymbol{I} = -\boldsymbol{D}^{-\frac{1}{2}}\boldsymbol{A}\boldsymbol{D}^{-\frac{1}{2}}$, let $\Delta_{\max}$ denote the maximum degree. Then:*

- *(Eigenvalue bounds)Chung (1997) The eigenvalues satisfy :*

$$|\lambda_i(\boldsymbol{L})| \leq 1 \quad while \quad \sqrt{\Delta_{\max}} \leq |\lambda_i(\boldsymbol{A})| \leq \Delta_{\max}$$

- *(Spectral radius) The spectral radius (largest absolute eigenvalue) satisfies:*

$$\rho(\boldsymbol{S}) \leq 1 \quad while \quad \rho(\boldsymbol{A}) \leq \Delta_{\max}$$

- *(Implications for diffusion) This boundedness ensures:*
    - *Uniform signal decay independent of graph degree distribution*
    - *Stable score function regardless of hub nodes*
    - *Consistent diffusion dynamics across heterogeneous graphs*

**Theorem 5.2** (Node Permutation Invariance of ESGD). *Let $G = (\boldsymbol{X}, \boldsymbol{A})$ be an undirected graph with adjacency $\boldsymbol{A}$ and node features $\boldsymbol{X}$. For any permutation matrix $\boldsymbol{P}$, define $\boldsymbol{X}' = \boldsymbol{P}\boldsymbol{X}$ and $\boldsymbol{A}' = \boldsymbol{P}\boldsymbol{A}\boldsymbol{P}^{\top}$. Let $\boldsymbol{S} = -\boldsymbol{D}^{-1/2}\boldsymbol{A}\boldsymbol{D}^{-1/2}$ and $\boldsymbol{S}' = -\boldsymbol{D}'^{-1/2}\boldsymbol{A}'\boldsymbol{D}'^{-1/2}$. Then the forward and reverse diffusion processes satisfy*

$$(\boldsymbol{X}'_t, \boldsymbol{\Lambda}'_t) \stackrel{d}{=} (\boldsymbol{P}\boldsymbol{X}_t, \boldsymbol{\Lambda}_t), \quad \boldsymbol{A}'_0 = \boldsymbol{P}\boldsymbol{A}_0\boldsymbol{P}^{\top},$$

The permutation invariance established above ensures that ESGD's learned distribution is well-defined over graph isomorphism classes. We now turn to the information-theoretic and optimization advantages of the SNL domain.

**Theorem 5.3** (Spectral SNR and Information Retention). *Let $\boldsymbol{X}_0 \in \mathbb{R}^n$ be the spectral embedding of a graph. Consider the forward process $\boldsymbol{X}_t = \sqrt{\bar{\alpha}_t}\boldsymbol{X}_0 + \sigma_t\varepsilon$, $\varepsilon \sim \mathcal{N}(\boldsymbol{0}, \boldsymbol{I}_n)$, with $\rho_t = \bar{\alpha}_t/\sigma_t^2$. Then:*

1. *(**SNR bound**) For any fixed initial data $\boldsymbol{x}_0$,*

$$\mathrm{SNR}(t) \leq \begin{cases} \rho_t, & \text{SNL domain } \boldsymbol{S}, \\ \Delta_{\max}^2 \rho_t, & \text{adjacency domain } \boldsymbol{A}. \end{cases}$$

2. *(**Mutual information**) For different initial data $\boldsymbol{X}_0$ is random with covariance $\boldsymbol{\Sigma}_0$, then*

$$I(\boldsymbol{X}_0; \boldsymbol{X}_t) \leq \tfrac{1}{2}\log\det(\boldsymbol{I} + \rho_t\boldsymbol{\Sigma}_0) \leq \tfrac{1}{2}\rho_t \mathbb{E}\|\boldsymbol{X}_0\|^2$$

*which scales as $O(\rho_t n)$ in domain $\boldsymbol{S}$ and $O(\rho_t n \Delta_{\max}^2)$ in domain $\boldsymbol{A}$.*

Theorem 5.3 establishes that the signal-to-noise ratio and mutual information in the SNL domain scale as $\mathcal{O}(\rho_t n)$, compared to $\mathcal{O}(\rho_t n \Delta_{\max}^2)$ in the adjacency domain. This $\Delta_{\max}^2$-factor reduction has three immediate consequences. First, for a fixed SNR target, the SNL formulation requires $\Delta_{\max}^2$ times fewer diffusion steps to achieve the same information retention in theorem 5.5, directly explaining the sampling acceleration observed in Table 6. Second, the bounded information flow ensures that gradient magnitudes remain stable across nodes of different degrees, preventing the hub-node dominance that plagues adjacency-based methods in theorem 5.4, 5.6 . Third, the uniform scaling allows a single set of hyper-parameters to work across diverse graph types, from regular grids to scale-free networks, without dataset-specific tuning. The following theorems quantify the stability implications through Lipschitz bounds, discretization error analysis and Fisher matrix.

**Theorem 5.4** (Score Lipschitz Bound). *For the score $s(\boldsymbol{x}, t) = \nabla_{\boldsymbol{x}} \log p_t(\boldsymbol{x})$ we have*

$$\|\nabla_{\boldsymbol{x}}s(\boldsymbol{x}, t)\|_{\mathrm{op}} \leq \sigma_t^{-2} + \frac{\bar{\alpha}_t}{\sigma_t^4} \cdot \frac{D_{\bullet}^2}{4},$$

*where $D_{\bullet}$ is the spectral diameter. Consequently,*

$$\|\nabla_{\boldsymbol{x}}s(\boldsymbol{x}, t)\|_{\mathrm{op}} \leq \begin{cases} \sigma_t^{-2} + \bar{\alpha}_t n/\sigma_t^4, & \boldsymbol{S}, \\ \sigma_t^{-2} + \bar{\alpha}_t n \Delta_{\max}^2/\sigma_t^4, & \boldsymbol{A}. \end{cases}$$

**Algorithm 1** Degree Matrix Recovery from $\boldsymbol{S}$

---

**Require:** SNL matrix $\boldsymbol{L}_{mod}$, threshold parameter $\delta > 0$
**Ensure:** Unweighted degree matrix $\boldsymbol{D}'$ and weighted adjacency matrix $\boldsymbol{A}$

1: **Step 1:** $\hat{\boldsymbol{A}}_{ij} \leftarrow \mathbf{1}_{|(\boldsymbol{S})_{ij}|>\delta}$ for all $(i,j)$          $\triangleright$ Identify graph structure by thresholding
2: **Step 2:** $d_i \leftarrow \sum_{j=1}^{n} \hat{\boldsymbol{A}}_{ij}$ for all $i$          $\triangleright$ Compute unweighted node degrees
3: **Step 3:** $\boldsymbol{D}' \leftarrow \mathrm{diag}(d_1, \ldots, d_n)$          $\triangleright$ Construct unweighted degree matrix
4: **Step 4 (For weighted graphs):** Recover the weighted adjacency matrix
5: **for all** $(i,j)$ with $\hat{\boldsymbol{A}}_{ij} = 1$ **do**          $\triangleright$ Edge weight recovery for connected pairs
6:      $\boldsymbol{A}_{ij} \leftarrow -(\boldsymbol{S})_{ij} \cdot \sqrt{d_i d_j}$
7: **end for**
8: $\boldsymbol{A}_{ij} \leftarrow 0$ for all $(i,j)$ with $\hat{\boldsymbol{A}}_{ij} = 0$          $\triangleright$ Zero weights for disconnected pairs
9: **return** $\boldsymbol{D}', \boldsymbol{A}$

---

**Theorem 5.5** (Drift Lipschitz and EM error). *The reverse-time SDE drift $b(\boldsymbol{x},t) = -\frac{1}{2}\beta(t)\boldsymbol{x} - \beta(t)s(\boldsymbol{x},t)$ has Lipschitz constant*

$$L_b(t) \ \leq \ \tfrac{1}{2}\beta(t) + \beta(t)\Big(\sigma_t^{-2} + \tfrac{\bar{\alpha}_t}{4\sigma_t^4}D_\bullet^2\Big).$$

*The Euler–Maruyama strong error satisfies*

$$\big(\mathbb{E}\|\boldsymbol{X}^{\mathrm{EM}} - \boldsymbol{X}\|^2\big)^{1/2} \leq C_{\mathrm{EM}}\Big(\int_0^1 L_b(t)^2 dt\Big)^{1/2}\Delta t^{1/2}.$$

**Theorem 5.6** (Fisher spectrum and conditioning). *Let $\boldsymbol{F} = \mathbb{E}[\nabla_{\boldsymbol{\theta}}\ell \nabla_{\boldsymbol{\theta}}\ell^\top]$ be the Fisher matrix of the score matching loss. Assume the network Jacobian satisfies $\|\boldsymbol{J}_{\boldsymbol{\theta}}(\boldsymbol{x},t)\| \leq C_{\mathrm{net}}(t)\|\boldsymbol{x}\|$ for all $\boldsymbol{x}, t$. Then:*

1. *(**Spectral bound**) The largest eigenvalue scales as*

$$\lambda_{\max}(\boldsymbol{F}) = \begin{cases} O(n), & \text{SNL domain,} \\ O(n\Delta_{\max}^2), & \text{adjacency domain.} \end{cases}$$

2. *(**Condition number**) If in addition $\lambda_{\min}(\boldsymbol{F}) \geq \gamma > 0$, then*

$$\kappa(\boldsymbol{F}) = \frac{\lambda_{\max}(\boldsymbol{F})}{\lambda_{\min}(\boldsymbol{F})} = \begin{cases} O(n/\gamma), & \boldsymbol{S}, \\ O(n\Delta_{\max}^2/\gamma), & \boldsymbol{A}. \end{cases}$$

## 6 GRAPH GENERATION RESULTS

### 6.1 GENERIC GRAPH GENERATION

**Datasets:** We test ESGD on *Community-small*, *Enzymes*, *Grid*, *Ego-small*, *Tree*, *Sbm*, and *Planar*. More details of these datasets are provided in Appendix C.1.

**Metrics:** We evaluate the maximum mean discrepancy (MMD) between equal numbers of generated and test graphs by measuring degree, clustering coefficient, 4-node orbit occurrences, their average and spectral in Table 1 and Table 2. See Appendix C.3 for details.

**Performance Analysis Across Graph Types**: Our experimental results reveal a systematic relationship between graph spectral properties and model performance. On datasets with bounded eigenvalue magnitudes (Tree at 3.05, Grid at 3.94), most baseline methods achieve reasonable performance. However, when eigenvalues increase to Enzymes at 5.30, Community-small at 6.61, Planar at 6.12, Ego-small at 9.04 and SBM at 14.13, adjacency-based methods (GSDM, GGSD) exhibit marked degradation due to spectral scaling as $(\Delta_{max}^2)$, where high-degree nodes dominate the training signal while low-degree nodes receive insufficient gradient updates.

**Evaluation Metric Hierarchy and Sensitivity Mechanisms**: This degradation is particularly evident in degree and clustering coefficient metrics, which we can explain through the hierarchical

Table 1: Generic graph generation on Community-small, Enzymes, Grid, and Ego-small. * The results were obtained by executing the published source code. Other results are taken from the published papers Luo et al. (2024); Wen et al. (2024); Jang et al. (2024); Eijkelboom et al. (2024). Hyphen (-) denotes that results are not provided and were not applicable due to memory issues. The best results are highlighted in **bold**, and the underline denotes the second best. Due to page limit we provide the standard deviations in Appendix E.1 and old models before 2022

| | Community-small | | | | Enzymes | | | | Grid | | | | Ego-small | | | |
| | Synthetic, ($12 \leq V \leq 20$) | | | | Real, ($10 \leq V \leq 125$) | | | | Synthetic, ($100 \leq V \leq 400$) | | | | Real, ($4 \leq V \leq 18$) | | | |
| | Deg.↓ | Clus.↓ | Orbit↓ | Avg.↓ | Deg.↓ | Clus.↓ | Orbit↓ | Avg.↓ | Deg.↓ | Clus.↓ | Orbit↓ | Avg.↓ | Deg.↓ | Clus.↓ | Orbit↓ | Avg.↓ |
|---|---|---|---|---|---|---|---|---|---|---|---|---|---|---|---|---|
| WSGM Guth et al. (2022) | 0.039 | 0.084 | 0.009 | 0.044 | 0.034 | 0.097 | 0.013 | 0.048 | 0.083 | 0.006 | 0.065 | 0.051 | - | - | - | - |
| GDSS Jo et al. (2022) | 0.045 | 0.086 | 0.007 | 0.046 | 0.026 | 0.102 | 0.009 | 0.046 | 0.111 | 0.005 | 0.070 | 0.062 | 0.021 | 0.024 | 0.007 | 0.017 |
| HGDM Wen et al. (2024) | 0.014 | 0.050 | 0.005 | 0.024 | 0.045 | 0.049 | 0.003 | 0.032 | 0.137 | 0.004 | 0.048 | 0.063 | 0.015 | 0.023 | 0.003 | 0.014 |
| GSDM * Luo et al. (2024) | 0.016 | 0.027 | 0.004 | 0.020 | 0.098 | 0.091 | 0.085 | 0.091 | 0.001 | 0.000 | 0.000 | 0.000 | 0.027 | 0.034 | 0.004 | 0.023 |
| GEEL Jang et al. (2024) | - | - | - | - | 0.005 | 0.018 | 0.006 | 0.010 | 0.000 | 0.000 | 0.000 | 0.000 | - | - | - | - |
| CatFlow Eijkelboom et al. (2024) | 0.018 | 0.086 | 0.007 | 0.037 | - | - | - | - | - | - | - | - | 0.013 | 0.024 | 0.008 | 0.015 |
| GGSD* Minello et al. (2025) | 0.027 | 0.082 | 0.011 | 0.040 | - | - | - | - | - | - | - | - | - | - | - | - |
| ESGD (ours) | 0.005 | 0.006 | 0.000 | 0.004 | 0.005 | 0.026 | 0.003 | 0.011 | 0.000 | 0.000 | 0.000 | 0.000 | 0.005 | 0.021 | 0.002 | 0.009 |

Table 2: Generic graph generation on Planar, SBM, and Tree. Results are taken from the published papers Jang et al. (2024); QIN et al. (2025); Bergmeister et al. (2024b).

| | Planar | | | | SBM | | | | Tree | | | |
| | Synthetic, ($|V| = 64$) | | | | Synthetic, ($31 \leq V \leq 187$) | | | | Synthetic, ($|V| = 64$) | | | |
| | Deg.↓ | Clus.↓ | Orbit↓ | Spec.↓ | Deg.↓ | Clus.↓ | Orbit↓ | Spec.↓ | Deg.↓ | Clus.↓ | Orbit↓ | Spec.↓ |
|---|---|---|---|---|---|---|---|---|---|---|---|---|
| SPECTRE Martinkus et al. (2022a) | 0.0005 | 0.0785 | 0.0012 | 0.0112 | 0.0015 | 0.0521 | 0.0412 | 0.0056 | - | - | - | - |
| DiGress Vignac et al. (2023a) | 0.0007 | 0.0780 | 0.0079 | 0.0098 | 0.0018 | 0.0485 | 0.0415 | 0.0045 | 0.2678 | 0.0428 | 0.0097 | 0.0123 |
| EDGE Chen et al. (2023b) | 0.0761 | 0.3229 | 0.7737 | 0.0957 | 0.0279 | 0.1113 | 0.0854 | 0.0251 | 0.0211 | 0.1207 | 0.0374 | 0.0438 |
| GDSS Jo et al. (2022) | 0.2500 | 0.3930 | 0.5870 | - | 0.4960 | 0.4560 | 0.7170 | - | - | - | - | - |
| GEEL* Jang et al. (2024) | 0.0006 | 0.0458 | 0.0000 | 0.0070 | 0.0034 | 0.0621 | 0.0000 | 0.0049 | - | - | - | - |
| DisCo Xu et al. (2024) | 0.0002 | 0.0403 | 0.0009 | - | 0.0006 | 0.0266 | 0.0510 | - | - | - | - | - |
| Cometh Siraudin et al. (2025) | 0.0006 | 0.0434 | 0.0016 | 0.0049 | 0.0020 | 0.0498 | 0.0383 | 0.0024 | - | - | - | - |
| DeFoG QIN et al. (2025) | 0.0005 | 0.0501 | 0.0006 | 0.0072 | 0.0006 | 0.0517 | 0.0556 | 0.0054 | 0.0002 | 0.0000 | 0.0000 | 0.0108 |
| Local PPGN (one-shot) Bergmeister et al. (2024b) | 0.0003 | 0.0245 | 0.0006 | 0.0104 | 0.0141 | 0.0528 | 0.0809 | 0.0071 | 0.0004 | 0.0000 | 0.0000 | 0.0080 |
| Local PPGN Bergmeister et al. (2024b) | 0.0005 | 0.0626 | 0.0017 | 0.0075 | 0.0119 | 0.0517 | 0.0669 | 0.0067 | 0.0001 | 0.0000 | 0.0000 | 0.0117 |
| GGSD* Minello et al. (2025) | 0.0024 | 0.0807 | 0.0048 | 0.0048 | 0.0041 | 0.0431 | 0.0730 | 0.0090 | - | - | - | - |
| ESGD (ours) | 0.0001 | 0.0228 | 0.0002 | 0.0057 | 0.0005 | 0.0027 | 0.0462 | 0.0039 | 0.0000 | 0.0001 | 0.0000 | 0.0081 |

structure of evaluation metrics and the population statistics of real-world graphs. Degree distribution operates as a macro-level indicator measuring node connectivity distributions. Clustering coefficient functions at the meso-level, quantifying neighborhood connection density. Orbit statistics capture micro-level patterns through four-node subgraph configurations. Real graphs typically exhibit power-law degree distributions where low-degree nodes comprise 70 to 80 percent of the population. For degree distribution, the numerical dominance of low-degree nodes makes their collective deviation the primary determinant of evaluation outcomes. For clustering coefficient, low-degree nodes exhibit heightened structural sensitivity: a degree-3 node shows clustering variations from 0 to 1.0 with single edge changes, whereas a degree-100 node requires hundreds of edges among 4950 possible neighbor connections to produce comparable variation.

**Theoretical Foundation for Empirical Performance**: ESGD's symmetric normalization directly addresses this challenge by eliminating eigenvalue weight bias, ensuring that nodes receive learning attention proportional to their population size rather than their degree. Since degree and clustering coefficient are dominated by the low-degree majority, ESGD's accurate modeling of this population translates directly into superior macro and meso-level performance, as confirmed by our experimental results in Tables 1 and 2. The moderate performance gap on orbit statistics reflects architectural boundaries: ESGD operates in global spectral space where the mapping to specific four-node configurations remains indirect, whereas explicit edge-level approaches like GEEL provide advantages for fine-grained motif detection. This trade-off is consistent with our design philosophy of prioritizing efficiency and scalability while maintaining competitive quality on the most population-representative metrics.

## 6.2 LARGE GRAPH GENERATION

**Datasets and Preprocessing:** We evaluate ESGD on three widely used citation networks: Cora (2708 nodes, 5429 edges), Citeseer (3312 nodes, 4715 edges), and PubMed (19717 nodes, 44338

edges). Beyond academic benchmarks, the ability to learn from single large graphs addresses critical industrial needs where data naturally exists as unified structures: social networks maintain billions of users in a single interconnected graph, enterprise knowledge graphs integrate all organizational entities into one coherent structure, and recommendation systems operate on unified user-item interaction networks. In each case, the generative model must extract patterns from one large network rather than learning from multiple independent instances.

Since spectral decomposition has quadratic complexity, we decompose each citation network into $k$-hop ego-subgraphs centered on individual nodes, where $k$ is chosen such that subgraphs contain 50 to 300 nodes on average. Each ego-subgraph preserves the $k$-hop neighborhood structure around its center node, capturing representative local motifs and degree patterns. This decomposition provides computational tractability while enabling the model to learn from multiple views of the original network structure. Detailed ego-subgraph statistics are provided in Table 9 and Appendix E.

**Metrics:** We use the same metrics as in Section 6.1.

Table 3: Large graph generation results on Cora, Citeseer, and PubMed. The baselines include diffusion-based generative models in the spectral space (SPECTRE, GSDM, GGSD) and discrete diffusion models (DisCo, Cometh, DeFoG).

| | Citeseer | | | | Cora | | | | PubMed | | | |
|---|---|---|---|---|---|---|---|---|---|---|---|---|
| | Deg.↓ | Clus.↓ | Orbit↓ | Avg.↓ | Deg.↓ | Clus.↓ | Orbit↓ | Avg.↓ | Deg.↓ | Clus.↓ | Orbit↓ | Avg.↓ |
| SPECTRE* Martinkus et al. (2022a) | 1.224 | 1.513 | 1.023 | 1.253 | 1.566 | 1.492 | 1.127 | 1.395 | 1.148 | 1.392 | 0.933 | 1.158 |
| GSDM* Luo et al. (2024) | 1.043 | 0.943 | 0.843 | 0.943 | 0.932 | 1.042 | 0.980 | 0.985 | 0.885 | 0.727 | 0.762 | 0.791 |
| GGSD* Minello et al. (2025) | 1.011 | 1.142 | 1.244 | 1.132 | 1.218 | 1.432 | 1.391 | 1.347 | 0.775 | 0.711 | 1.029 | 0.838 |
| DisCo* Xu et al. (2024) | 0.893 | 0.654 | 0.896 | 0.814 | 0.918 | 0.775 | 0.564 | 0.752 | 0.637 | 0.611 | 0.815 | 0.688 |
| Cometh* Siraudin et al. (2025) | 0.985 | 0.856 | 1.001 | 0.947 | 0.751 | 0.899 | 0.541 | 0.730 | 0.597 | 0.625 | 0.437 | 0.553 |
| DeFoG* QIN et al. (2025) | 0.496 | 0.656 | 0.910 | 0.671 | 0.758 | 0.756 | 0.501 | 0.672 | 0.355 | 0.496 | 0.308 | 0.386 |
| ESGD (ours) | **0.329** | **0.606** | **0.314** | **0.433** | **0.311** | **0.573** | **0.192** | **0.359** | **0.215** | **0.475** | **0.109** | **0.266** |

**Results and Analysis:** Table 3 shows that ESGD consistently outperforms all baselines on citation networks, achieving the best average MMD across all three datasets. We attribute ESGD's superior performance to its robustness against the inherent heterogeneity of ego-subgraphs. Unlike synthetic benchmarks with uniform structure, ego-subgraphs extracted from real-world citation networks exhibit highly non-uniform distributions. As shown in Figure 1b, the condition numbers of these subgraphs span multiple orders of magnitude, creating a challenging learning problem. SPECTRE suffers from training instability and convergence difficulties under such heterogeneous conditions, while GGSD fails to effectively capture the coupling between eigenvalues and eigenvectors. Discrete diffusion models require longer training schedules and larger model to accommodate outlier subgraphs with atypical properties. In contrast, ESGD's achieves strong performance with a lightweight architecture while maintaining structural fidelity across diverse local neighborhoods.

## 6.3 MOLECULES GENERATION

**Datasets:** We test ESGD on four molecule benchmarks: QM9 (Ramakrishnan et al., 2014), ZINC250k (Irwin et al., 2012), Moses (Polykovskiy et al., 2020), and GuacaMol (Brown et al., 2019).

**Metrics:** We evaluate the quality of 10,000 generated graphs using Frechet ChemNet Distance (FCD) (Preuer et al., 2018), Neighborhood Subgraph Pairwise Distance Kernel (NSPDK) MMD (Costa & De Grave, 2010), validity w/o correction abbreviated as Val. w/o, validity, and the generation time for QM9 and ZINC250k. For Moses, we additionally report uniqueness, novelty, filters, SNN (similarity to nearest neighbor), and scaffold similarity (Scaf). For GuacaMol, we report validity, valid & unique (V.U.), valid & unique & novel (V.U.N.), KL divergence, and FCD. Please see Appendix C.4 for more details.

As shown in Tables 4 and 5, ESGD achieves competitive performance across all molecular benchmarks. Notably, the Moses benchmark provides the most comprehensive assessment of whether a generative model truly learns data distributions rather than merely memorizing training samples, as it evaluates validity, uniqueness, novelty, and structural diversity simultaneously. ESGD achieves the highest validity of 94.6% on Moses while maintaining strong performance across all

Table 4: Results on the QM9 and ZINC250k. Results were taken from the published papers Luo et al. (2024); Wen et al. (2024); Jang et al. (2024); Eijkelboom et al. (2024); QIN et al. (2025). We provide the validity, uniqueness, and novelty values in Appendix E.2 due to page limit.

| | QM9 | | | | | ZINC250k | | | | |
|---|---|---|---|---|---|---|---|---|---|---|
| | Validity (%)↑ | Val. w/o (%)↑ | NSPDK↓ | FCD↓ | Time(s)↓ | Validity (%)↑ | Val. w/o (%)↑ | NSPDK↓ | FCD↓ | Time(s)↓ |
| GraphAF (Shi et al., 2020) | **100** | 67 | 0.020 | 5.268 | $2.28e^3$ | **100** | 68 | 0.044 | 16.289 | $5.72e^3$ |
| GraphAF+FC | **100** | 74.43 | 0.021 | 5.625 | $2.32e^3$ | **100** | 68.47 | 0.044 | 16.023 | $5.91e^3$ |
| GraphDF (Luo et al., 2021) | **100** | 82.67 | 0.063 | 10.816 | $5.08e^4$ | **100** | 89.03 | 0.176 | 34.202 | $5.87e^4$ |
| GraphDF+FC | **100** | 93.88 | 0.064 | 10.928 | $4.72e^4$ | **100** | 90.61 | 0.177 | 33.546 | $5.79e^4$ |
| MoFlow (Zang & Wang, 2020) | **100** | 91.36 | 0.017 | 4.467 | **4.58** | **100** | 63.11 | 0.046 | 20.931 | 25.9 |
| EDP-GNN (Niu et al., 2020a) | **100** | 47.52 | 0.005 | 2.680 | $4.13e^3$ | **100** | 82.97 | 0.049 | 16.737 | $8.41e^3$ |
| GDSS (Jo et al., 2022) | **100** | 95.72 | 0.019 | 2.900 | $1.06e^2$ | **100** | 97.01 | 0.019 | 14.656 | $2.11e^3$ |
| HGDM (Wen et al., 2024) | **100** | 98.04 | 0.002 | 2.131 | $1.23e^2$ | **100** | 93.51 | 0.016 | 17.69 | $2.23e^3$ |
| GSDM* (Luo et al., 2024) | **100** | 99.81 | 0.009 | 3.191 | 18.5 | **100** | 93.0 | 0.016 | 12.07 | 86.3 |
| GEEL Jang et al. (2024) | 100.0 | 100 | 0.0002 | 0.089 | - | **100** | 99.31 | 0.0068 | 0.401 | - |
| CatFlow Eijkelboom et al. (2024) | **100** | 99.81 | - | 0.441 | - | **100** | 99.21 | - | 13.211 | - |
| DeFog QIN et al. (2025) | - | - | - | - | - | **100** | 99.22 | 0.0008 | 1.425 | - |
| ESGD (ours) | **100** | 99.20 | 0.002 | 1.425 | 14.6 | **100** | 98.29 | 0.010 | 8.80 | 72.1 |

Table 5: Results on Moses and GuacaMol. Results were taken from the published papers Vignac et al. (2023a); Xu et al. (2024); Siraudin et al. (2025); QIN et al. (2025).

| | GuacaMol | | | | | Moses | | | | | | |
|---|---|---|---|---|---|---|---|---|---|---|---|---|
| | Val.↑ | V.U.↑ | V.U.N.↑ | KL div↑ | FCD↑ | Val.↑ | Uniq.↑ | Nov.↑ | Filters↑ | FCD↓ | SNN↑ | Scaf↑ |
| DiGress (Vignac et al., 2023a) | 85.2 | 85.2 | 85.1 | 92.9 | 68.0 | 85.7 | **100.0** | 95.0 | 97.1 | **1.19** | 0.52 | 14.8 |
| DisCo (Xu et al., 2024) | 86.6 | 86.6 | 86.5 | 92.6 | 59.7 | 88.3 | **100.0** | 97.7 | 95.6 | 1.44 | 0.50 | 15.1 |
| Cometh (Siraudin et al., 2025) | 98.9 | 98.9 | 97.6 | 96.7 | 72.7 | 90.5 | 99.9 | 92.6 | 99.1 | 1.27 | 0.54 | 16.0 |
| DeFoG (10% steps) (QIN et al., 2025) | 91.7 | 91.7 | 91.2 | 92.3 | 57.9 | 83.9 | 99.9 | 96.9 | 96.5 | 1.87 | 0.50 | **23.5** |
| DeFoG (QIN et al., 2025) | 99.0 | 99.0 | 97.9 | 97.7 | 73.8 | 92.8 | 99.9 | 92.1 | 98.9 | 1.95 | 0.55 | 14.4 |
| ESGD (ours) | **99.1** | **99.1** | 98.3 | 98.0 | 76.9 | 94.6 | 99.9 | 93.4 | 98.9 | 1.92 | **0.58** | 15.7 |

metrics, confirming that our spectral approach genuinely captures the underlying molecular distribution. These results demonstrate ESGD's effectiveness and generalization capability on complex molecular graphs with multiple node types and weighted edges.

## 6.4 EFFICIENCY EVALUATION

**Computational Efficiency:** We compare the computational efficiency of recent graph generation models on the *Planar* dataset, as it is widely adopted across most state-of-the-art methods and provides well-tuned configurations for fair comparison. The results are summarized in Table 6.

For a fair evaluation, we compute the parameter counts based on each model's reported configuration for the *Planar* dataset. FLOPs are calculated as the total computational cost required to generate a single graph with 64 nodes. For GAN-based models such as SPECTRE, we report only the generator's FLOPs, while for diffusion-based models, the total FLOPs are computed as: Total FLOPs = FLOPs per step × Number of sampling steps. The "Steps" column indicates the minimum number of sampling steps required for each model to achieve its best reported performance. For GAN and autoregressive models, this value is fixed at 1. We exclude models published before 2022 due to their significantly less competitive performance.

Table 6: Efficiency comparison on the *Planar* dataset. The best results are highlighted in **bold**.

| Method | Type | Parameter Counts | Training Epochs | FLOPs/Step | FLOPs |
|---|---|---|---|---|---|
| SPECTRE | GAN | 0.36M (Generator) | 12,000 | 2.27G | 2.27G (1 Step) |
| DiGress | Diffusion (Discrete) | 8.89M | 100,000 | 5.29G | 5.29T (1000 Steps) |
| GEEL | Autoregressive LSTM | 7.17M | 5,000 | 2.41G | 2.41G (1 Step) |
| DisCo | Diffusion (Discrete) | 4.49M | 50,000 | 2.68G | 134.12G (50 Steps) |
| Cometh | Diffusion (Continuous-Time Discrete) | 5.29M | 150,000 | 5.28G | 5.28T (1000 Steps) |
| DeFoG | Flow-Matching | 6.59M | 100,000 | 5.28G | 5.28T (1000 Steps) |
| Local PPGN (one-shot) | Diffusion (Discrete) | 3.73M | ∞ (steps-based) | 30.23G | 7.74T (256 Steps) |
| GGSD | Diffusion (Continuous) | 20.09M | ∼500K (batches) | 0.92G | 92.09G (100 Steps) |
| ESGD (ours) | Diffusion (Continuous) | **0.21M** | **1,000** | **45.2M** | **2.26G (50 Steps)** |

**Computational Implications.** The spectral compression strategy has direct consequences for model complexity and training efficiency. As we discussed in Section 6.1, adjacency-based spectral models suffer from eigenvalue imbalance and condition number outliers, requiring large model capacity to handle the resulting variance. Discrete diffusion models face complementary challenges: they must maintain large parameter counts to model complex categorical transitions over edge states, require extensive training schedules spanning hundreds of thousands of iterations, and demand long sampling chains with hundreds to thousands of denoising steps to achieve high-quality generation.

ESGD addresses both limitations through its spectral compression design. First, by bounding all eigenvalues to the interval $[-1, 1]$, we eliminate degree-dependent scaling that force adjacency-based methods to allocate substantial model capacity for counteracting spectral variance. Second, by fixing the eigenvectors $\mathbf{U}$ and diffusing only the eigenvalues $\mathbf{\Lambda}$, we drastically reduce the effective function class that the score network must approximate. The eigenvalue score network $s_\phi$ operates on an $n$-dimensional vector rather than an $n \times n$ matrix. These architectural simplifications achieves competitive generation quality with substantially fewer parameters than existing models.

Table 6 quantifies these efficiency gains on the Planar dataset. ESGD achieves substantial improvements in both parameter efficiency and computational cost compared to all baseline methods. With only **0.21M** parameters and **2.26G** total FLOPs per generated graph, ESGD attains comparable or superior generation quality while matching the efficiency of single-step models such as SPEC-TRE and GEEL. Compared to diffusion-based baselines, ESGD reduces computational cost by over **2000×**. This efficiency advantage stems directly from our spectral normalization strategy, which, as discussed in Sections 3.1 and 5, transforms heterogeneous graph distributions into well-conditioned representations that can be modeled with compact neural architectures. Most existing diffusion approaches allocate large model capacity to counteract the variance introduced by unnormalized spectra, whereas ESGD's bounded spectral domain allows the score network to focus its limited capacity on learning the essential structural patterns.

**Sampling Steps:** A key factor contributing to ESGD's computational efficiency is its ability to achieve high-quality generation with significantly fewer sampling steps compared to other diffusion-based methods. As shown in Table 6, while most diffusion models require hundreds to thousands of steps (e.g., DiGress, Cometh, and DeFoG with 1000 steps; Local PPGN with 256 steps), ESGD achieves optimal performance with only **50 steps**. Even GGSD, which also operates in the spectral domain, requires 100 steps to achieve comparable quality.

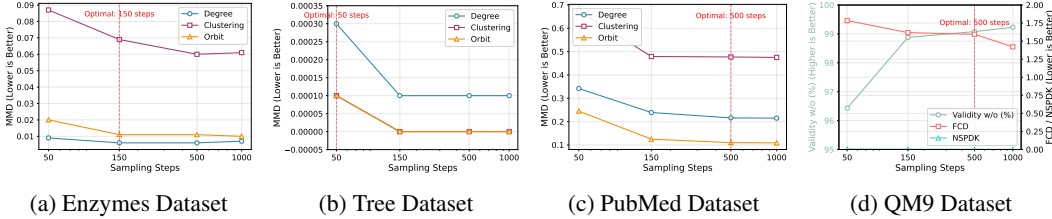

| (a) Enzymes Dataset | (b) Tree Dataset | (c) PubMed Dataset | (d) QM9 Dataset |

Figure 3: The sub-figures show how evaluation metrics change with different sampling steps. Red dashed lines indicate the optimal sampling steps for each dataset.

Figure 3 further illustrates how ESGD's performance metrics stabilize rapidly across diverse datasets. On the Enzymes dataset, the MMD metrics converge by 150 steps, while on the Tree dataset, near-optimal performance is achieved with just 50 steps. For molecular graphs (QM9), ESGD achieves 99.08% validity at 500 steps, demonstrating excellent efficiency without sacrificing generation quality.

## 7 CONCLUSIONS

We have presented a spectral perspective on graph diffusion that achieves both theoretical soundness and practical efficiency. The broader lesson is that progress in generative modeling may not always come from additional layers of engineering, but from revisiting the core formulations that govern stability and scalability. Subtle adjustments to these foundations can sometimes prove more effective than increasingly intricate designs, a direction our work illustrates for graph generation.

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

## A  THEORETICAL SUPPORT

**Theorem A.1** (Permutation invariance). *Let $G = (\boldsymbol{X}, \boldsymbol{A})$ be an undirected graph with adjacency $\boldsymbol{A}$ and node features $\boldsymbol{X}$. For any permutation matrix $\boldsymbol{P}$, set $\boldsymbol{X}' = \boldsymbol{P}\boldsymbol{X}$ and $\boldsymbol{A}' = \boldsymbol{P}\boldsymbol{A}\boldsymbol{P}^\top$. Let $\boldsymbol{S} = -\boldsymbol{D}^{-1/2}\boldsymbol{A}\boldsymbol{D}^{-1/2}$ and $\boldsymbol{S}' = -\boldsymbol{D}'^{-1/2}\boldsymbol{A}'\boldsymbol{D}'^{-1/2}$. Then the ESGD forward and reverse diffusion processes satisfy*

$$(\boldsymbol{X}'_t, \boldsymbol{\Lambda}'_t) \stackrel{d}{=} (\boldsymbol{P}\boldsymbol{X}_t, \boldsymbol{\Lambda}_t), \qquad \boldsymbol{A}'_0 = \boldsymbol{P}\boldsymbol{A}_0\boldsymbol{P}^\top,$$

*so the generative distribution is invariant to node permutations.*

*Proof.* **(1) Operator similarity.** Since $\boldsymbol{D}' = \mathrm{diag}(\boldsymbol{A}'\boldsymbol{1}) = \mathrm{diag}(\boldsymbol{P}\boldsymbol{A}\boldsymbol{P}^\top\boldsymbol{1}) = \boldsymbol{P}\boldsymbol{D}\boldsymbol{P}^\top$ and $\boldsymbol{P}$ is orthogonal, we have

$$\boldsymbol{S}' = -\boldsymbol{D}'^{-1/2}\boldsymbol{A}'\boldsymbol{D}'^{-1/2} = -(\boldsymbol{P}\boldsymbol{D}\boldsymbol{P}^\top)^{-1/2}(\boldsymbol{P}\boldsymbol{A}\boldsymbol{P}^\top)(\boldsymbol{P}\boldsymbol{D}\boldsymbol{P}^\top)^{-1/2} = \boldsymbol{P}\boldsymbol{S}\boldsymbol{P}^\top.$$

Hence $\boldsymbol{S}$ and $\boldsymbol{S}'$ are similar and share eigenvalues; their eigenvectors transform as $\boldsymbol{U}' = \boldsymbol{P}\boldsymbol{U}$ (e.g., (Chung, 1997, Ch. 1)). **(2) Equivariance of message passing layers.** Consider a standard (normalized) GCN/MPNN layer

$$\Phi(\boldsymbol{H}; \boldsymbol{S}) = \sigma(-\boldsymbol{S}\boldsymbol{H}\boldsymbol{W}),$$

with elementwise activation $\sigma$ and weight matrix $\boldsymbol{W}$. Using $\boldsymbol{P}\boldsymbol{H}\boldsymbol{W} = (\boldsymbol{P}\boldsymbol{H})\boldsymbol{W}$ and $\boldsymbol{P}\boldsymbol{S}\boldsymbol{P}^\top\boldsymbol{P}\boldsymbol{H} = \boldsymbol{P}(-\boldsymbol{S}\boldsymbol{H}\boldsymbol{W})$,

$$\Phi(\boldsymbol{P}\boldsymbol{H}; \boldsymbol{P}\boldsymbol{S}\boldsymbol{P}^\top) = \sigma\big(-(\boldsymbol{P}\boldsymbol{S}\boldsymbol{P}^\top)(\boldsymbol{P}\boldsymbol{H})\boldsymbol{W}\big) = \boldsymbol{P}\,\sigma(-\boldsymbol{S}\boldsymbol{H}\boldsymbol{W}) = \boldsymbol{P}\,\Phi(\boldsymbol{H}; \boldsymbol{S}).$$

Thus each layer is permutation-equivariant; stacked networks and the score nets inherit equivariance (see also Zaheer et al. (2017); Xu et al. (2019)). **(3) Forward SDE equivariance.** The forward SDEs read

$$\mathrm{d}\boldsymbol{X}_t = f_{\boldsymbol{X}}(\boldsymbol{X}_t, \boldsymbol{\Lambda}_t, t; \boldsymbol{S})\,\mathrm{d}t + g_{\boldsymbol{X}}(t)\,\mathrm{d}\boldsymbol{W}_t^{(\boldsymbol{X})}, \quad \mathrm{d}\boldsymbol{\Lambda}_t = f_{\boldsymbol{\Lambda}}(\boldsymbol{\Lambda}_t, t)\,\mathrm{d}t + g_{\boldsymbol{\Lambda}}(t)\,\mathrm{d}\boldsymbol{W}_t^{(\boldsymbol{\Lambda})}.$$

Define $\boldsymbol{X}'_t = \boldsymbol{P}\boldsymbol{X}_t$, $\boldsymbol{\Lambda}'_t = \boldsymbol{\Lambda}_t$, and $\boldsymbol{W}_t^{(\boldsymbol{X})'} = \boldsymbol{P}\boldsymbol{W}_t^{(\boldsymbol{X})}$. Since Brownian motion is invariant under orthogonal transforms and $f_{\boldsymbol{X}}(\cdot; \boldsymbol{S})$ is permutation-equivariant by (2), we obtain

$$\mathrm{d}\boldsymbol{X}'_t = f_{\boldsymbol{X}}(\boldsymbol{X}'_t, \boldsymbol{\Lambda}'_t, t; \boldsymbol{S}')\,\mathrm{d}t + g_{\boldsymbol{X}}(t)\,\mathrm{d}\boldsymbol{W}_t^{(\boldsymbol{X})'}.$$

Therefore the forward process is permutation-equivariant (e.g., (Øksendal, 2003, Ch. 3)).
**(4) Scores and reverse SDE.** Let $p_t$ be the joint density of $(\boldsymbol{X}_t, \boldsymbol{\Lambda}_t)$. For any permutation $\boldsymbol{P}$, $p_t^{(\boldsymbol{P})}(\boldsymbol{x}, \lambda) = p_t(\boldsymbol{P}^\top\boldsymbol{x}, \lambda)$

$$\Rightarrow \quad \nabla_{\boldsymbol{x}} \log p_t(\boldsymbol{P}\boldsymbol{x}, \lambda) = \boldsymbol{P}\,\nabla_{\boldsymbol{x}} \log p_t(\boldsymbol{x}, \lambda), \;\; \nabla_\lambda \log p_t(\boldsymbol{P}\boldsymbol{x}, \lambda) = \nabla_\lambda \log p_t(\boldsymbol{x}, \lambda).$$

Hence the ground-truth scores are permutation-equivariant and so are consistent score networks trained by score matching. The reverse-time SDE (variance-exploding case) is

$$\mathrm{d}\boldsymbol{X}_t = \big(-\tfrac{1}{2}\beta(t)\boldsymbol{X}_t - \beta(t)\,s(\boldsymbol{X}_t, t)\big)\,\mathrm{d}t + \sqrt{\beta(t)}\,\mathrm{d}\bar{\boldsymbol{W}}_t,$$

which remains permutation-equivariant when replacing $(\boldsymbol{X}_t, s)$ by $(\boldsymbol{P}\boldsymbol{X}_t, \boldsymbol{P}s)$.
**(5) Reconstruction.** At termination, $(\boldsymbol{X}_0, \boldsymbol{\Lambda}_0) \mapsto (\boldsymbol{P}\boldsymbol{X}_0, \boldsymbol{\Lambda}_0)$ and

$$\boldsymbol{S}'_0 = (\boldsymbol{P}\boldsymbol{U}_0)\boldsymbol{\Lambda}_0(\boldsymbol{P}\boldsymbol{U}_0)^\top = \boldsymbol{P}\boldsymbol{S}_0\boldsymbol{P}^\top.$$

With $\boldsymbol{A}_0 = -\boldsymbol{D}^{1/2}\boldsymbol{S}_0\boldsymbol{D}^{1/2}$ and $\boldsymbol{A}'_0 = -\boldsymbol{D}'^{1/2}\boldsymbol{S}'_0\boldsymbol{D}'^{1/2}$, and $\boldsymbol{D}' = \boldsymbol{P}\boldsymbol{D}\boldsymbol{P}^\top$, we get $\boldsymbol{A}'_0 = \boldsymbol{P}\boldsymbol{A}_0\boldsymbol{P}^\top$. This proves the claim. $\qquad\square$

Let $\boldsymbol{S}_t = \boldsymbol{U}_0 \boldsymbol{\Lambda}_t \boldsymbol{U}_0^\top$ with fixed $\boldsymbol{U}_0$ chosen once by eigendecomposition. For any block-orthogonal rotation $\boldsymbol{R}$ acting within degenerate eigenspaces of $\boldsymbol{U}_0$, set $\boldsymbol{U}_0' = \boldsymbol{U}_0 \boldsymbol{R}$. Then

$$\boldsymbol{U}_0' \boldsymbol{\Lambda}_t \boldsymbol{U}_0'^\top = \boldsymbol{U}_0 \boldsymbol{R} \boldsymbol{\Lambda}_t \boldsymbol{R}^\top \boldsymbol{U}_0^\top = \boldsymbol{U}_0 \boldsymbol{\Lambda}_t \boldsymbol{U}_0^\top,$$

so the reconstructed operator and hence the generated distribution are independent of the particular basis within degenerate subspaces (cf. von Luxburg (2007)).

**Definition A.2** (Spectral diameter). Let $\mathbb{X}_\bullet \subset \mathbb{R}^n$ be the feasible spectral set in domain $\bullet \in \{\boldsymbol{S}, \boldsymbol{A}\}$ (SNL $\boldsymbol{S}$ or adjacency $\boldsymbol{A}$). There exist absolute constants

$$D_{\boldsymbol{S}} = 2\sqrt{n}, \qquad D_{\boldsymbol{A}} = 2\Delta_{\max}\sqrt{n}$$

such that any spectral embedding $\boldsymbol{x}_0 \in \mathbb{X}_\bullet$ satisfies $\|\boldsymbol{x}_0\|^2 \le D_\bullet^2/4$. For $\boldsymbol{S}$, this follows from the spectrum lying in $[-1, 1]$; for $\boldsymbol{A}$, from $\|\boldsymbol{A}\| \le \Delta_{\max}$ (see Chung (1997)).

**Theorem A.3** (Spectral SNR and information retention). *Let $\boldsymbol{X}_0 \in \mathbb{R}^n$ be a spectral embedding. Consider*

$$\boldsymbol{X}_t = \sqrt{\bar{\alpha}_t}\,\boldsymbol{X}_0 + \sigma_t \varepsilon, \qquad \varepsilon \sim \mathcal{N}(\boldsymbol{0}, \boldsymbol{I}_n), \qquad \rho_t := \bar{\alpha}_t / \sigma_t^2.$$

*Then:*

   1. *(**SNR bound**) For any fixed $\boldsymbol{x}_0$,*

$$\mathrm{SNR}(t) := \frac{\bar{\alpha}_t \|\boldsymbol{x}_0\|^2}{n\sigma_t^2} \le \begin{cases} \rho_t, & \bullet = \boldsymbol{S}, \\ \Delta_{\max}^2 \rho_t, & \bullet = \boldsymbol{A}. \end{cases}$$

   2. *(**Mutual information**) If $\boldsymbol{X}_0$ has covariance $\boldsymbol{\Sigma}_0$, then*

$$I(\boldsymbol{X}_0; \boldsymbol{X}_t) \le \tfrac{1}{2} \log \det(\boldsymbol{I} + \rho_t \boldsymbol{\Sigma}_0) \le \tfrac{1}{2}\rho_t \,\mathbb{E}\|\boldsymbol{X}_0\|^2,$$

   *with $\mathbb{E}\|\boldsymbol{X}_0\|^2 = O(n)$ in domain $\boldsymbol{S}$ and $O(n\Delta_{\max}^2)$ in domain $\boldsymbol{A}$.*

*Proof.* (1) By Definition A.2, $\|\boldsymbol{x}_0\|^2 \le D_\bullet^2/4$, hence

$$\mathrm{SNR}(t) = \rho_t \frac{\|\boldsymbol{x}_0\|^2}{n} \le \rho_t \cdot \frac{D_\bullet^2}{4n} = \begin{cases} \rho_t, & D_{\boldsymbol{S}}^2/4 = n, \\ \Delta_{\max}^2\rho_t, & D_{\boldsymbol{A}}^2/4 = n\Delta_{\max}^2. \end{cases}$$

(2) Since $\boldsymbol{X}_t = \sqrt{\bar{\alpha}_t}\boldsymbol{X}_0 + \sigma_t \varepsilon$ with $\varepsilon \perp \boldsymbol{X}_0$, the Gaussian channel formula gives $I(\boldsymbol{X}_0; \boldsymbol{X}_t) = \tfrac{1}{2} \log \det(\boldsymbol{I} + \rho_t \boldsymbol{\Sigma}_0)$ (e.g., (Cover & Thomas, 2006, Ch. 9)). Using $\log \det(\boldsymbol{I} + \boldsymbol{M}) \le \mathrm{tr}(\boldsymbol{M})$ for $\boldsymbol{M} \succeq \boldsymbol{0}$,

$$I(\boldsymbol{X}_0; \boldsymbol{X}_t) \le \tfrac{1}{2} \mathrm{tr}(\rho_t \boldsymbol{\Sigma}_0) = \tfrac{1}{2}\rho_t \,\mathbb{E}\|\boldsymbol{X}_0\|^2.$$

By Definition A.2, any $\boldsymbol{X}_0$ supported on $\mathbb{X}_\bullet$ satisfies $\mathbb{E}\|\boldsymbol{X}_0\|^2 \le D_\bullet^2/4$, which yields the stated domain scalings. $\qquad\square$

**Theorem A.4** (Score Lipschitz). *Let $p_t$ be the density of $\boldsymbol{X}_t = \sqrt{\bar{\alpha}_t}\boldsymbol{X}_0 + \sigma_t \varepsilon$ with $\varepsilon \sim \mathcal{N}(\boldsymbol{0}, \boldsymbol{I}_n)$ and define the score $s(\boldsymbol{x}, t) = \nabla_{\boldsymbol{x}} \log p_t(\boldsymbol{x})$. Then*

$$\|\nabla_{\boldsymbol{x}} s(\boldsymbol{x}, t)\|_{\mathrm{op}} \le \sigma_t^{-2} + \frac{\bar{\alpha}_t}{\sigma_t^4} \cdot \frac{D_\bullet^2}{4} \le \begin{cases} \sigma_t^{-2} + \bar{\alpha}_t n/\sigma_t^4, & \bullet = \boldsymbol{S}, \\ \sigma_t^{-2} + \bar{\alpha}_t n\Delta_{\max}^2/\sigma_t^4, & \bullet = \boldsymbol{A}. \end{cases}$$

*Proof.* Fix $t$ and write $\bar{\alpha} = \bar{\alpha}_t$, $\sigma = \sigma_t$. Denote $m(\boldsymbol{x}) := \mathbb{E}[\boldsymbol{X}_0 \mid \boldsymbol{X} = \boldsymbol{x}]$. By Tweedie's identity for additive Gaussian noise (see Efron (2011) and also (Song et al., 2021b, Sec. 3)),

$$\sqrt{\bar{\alpha}}\, m(\boldsymbol{x}) = \boldsymbol{x} + \sigma^2 s(\boldsymbol{x}, t) \iff s(\boldsymbol{x}, t) = \frac{\sqrt{\bar{\alpha}}\, m(\boldsymbol{x}) - \boldsymbol{x}}{\sigma^2}. \tag{4}$$

Differentiating equation 4 in $\boldsymbol{x}$ yields

$$\nabla s(\boldsymbol{x}, t) = \frac{\sqrt{\bar{\alpha}}}{\sigma^2} \nabla m(\boldsymbol{x}) - \frac{1}{\sigma^2}\boldsymbol{I}. \tag{5}$$

For the Gaussian corruption channel $\boldsymbol{X} = \sqrt{\bar{\alpha}}\boldsymbol{X}_0 + \sigma \varepsilon$, a standard covariance identity (Stein's lemma / Bayes rule differentiation) gives

$$\nabla m(\boldsymbol{x}) = \frac{\sqrt{\bar{\alpha}}}{\sigma^2} \mathrm{Cov}(\boldsymbol{X}_0 \mid \boldsymbol{X} = \boldsymbol{x}), \tag{6}$$

see, e.g., (Efron, 2011, Sec. 2). Substituting equation 6 into equation 5,

$$\nabla s(\boldsymbol{x}, t) = \frac{\bar{\alpha}}{\sigma^4} \operatorname{Cov}(\boldsymbol{X}_0 \mid \boldsymbol{X} = \boldsymbol{x}) - \frac{1}{\sigma^2} \boldsymbol{I}. \tag{7}$$

Since $\boldsymbol{X}_0 \in \mathbb{X}_\bullet$ almost surely, any conditional distribution $\boldsymbol{X}_0 \mid \boldsymbol{X} = \boldsymbol{x}$ is supported on $\mathbb{X}_\bullet$. Hence, by a diameter (Popoviciu-type) bound for bounded random vectors,

$$\|\operatorname{Cov}(\boldsymbol{X}_0 \mid \boldsymbol{X} = \boldsymbol{x})\|_{\mathrm{op}} \leq \frac{D_\bullet^2}{4}.$$

Taking operator norms in equation 7 and using the triangle inequality yields

$$\|\nabla s(\boldsymbol{x}, t)\|_{\mathrm{op}} \leq \frac{1}{\sigma^2} + \frac{\bar{\alpha}}{\sigma^4} \cdot \frac{D_\bullet^2}{4},$$

and substituting $D_{\boldsymbol{S}}^2/4 = n$ and $D_{\boldsymbol{A}}^2/4 = n\Delta_{\max}^2$ completes the proof. $\square$

**Theorem A.5** (Drift Lipschitz and EM error). *Consider the reverse-time SDE in variance-exploding form*

$$\mathrm{d}\boldsymbol{X}_t = \underbrace{\left( -\tfrac{1}{2}\beta(t)\boldsymbol{X}_t - \beta(t)s(\boldsymbol{X}_t, t) \right)}_{=: b(\boldsymbol{X}_t, t)} \mathrm{d}t + \sqrt{\beta(t)}\, \mathrm{d}\bar{\boldsymbol{W}}_t.$$

*Then, for each $t$,*

$$L_b(t) := \sup_{\boldsymbol{x}} \|\nabla_{\boldsymbol{x}} b(\boldsymbol{x}, t)\|_{\mathrm{op}} \leq \tfrac{1}{2}\beta(t) + \beta(t)\left( \sigma_t^{-2} + \frac{\bar{\alpha}_t}{4\sigma_t^4} D_\bullet^2 \right).$$

*Moreover, the Euler–Maruyama (EM) strong error with step size $\Delta t$ satisfies*

$$\left( \mathbb{E}\|\boldsymbol{X}_1^{\mathrm{EM}} - \boldsymbol{X}_1\|^2 \right)^{1/2} \leq C_{\mathrm{EM}} \left( \int_0^1 L_b(t)^2\, \mathrm{d}t \right)^{1/2} \Delta t^{1/2}.$$

*Proof.* The Jacobian of $b(\cdot, t)$ is

$$\nabla_{\boldsymbol{x}} b(\boldsymbol{x}, t) = -\tfrac{1}{2}\beta(t)\boldsymbol{I} - \beta(t)\nabla_{\boldsymbol{x}} s(\boldsymbol{x}, t).$$

Hence

$$\|\nabla_{\boldsymbol{x}} b(\boldsymbol{x}, t)\|_{\mathrm{op}} \leq \tfrac{1}{2}\beta(t) + \beta(t)\,\|\nabla_{\boldsymbol{x}} s(\boldsymbol{x}, t)\|_{\mathrm{op}}.$$

Applying Theorem A.4 gives the bound on $L_b(t)$. For EM, consider the time-inhomogeneous SDE $\mathrm{d}\boldsymbol{X}_t = b(\boldsymbol{X}_t, t)\, \mathrm{d}t + \sigma(t)\, \mathrm{d}\bar{\boldsymbol{W}}_t$ with $\sigma(t) = \sqrt{\beta(t)}\boldsymbol{I}$ independent of $\boldsymbol{x}$. Under global $\boldsymbol{x}$-Lipschitz continuity of $b(\cdot, t)$ with modulus $L_b(t)$ and linear growth (both satisfied here), the classical EM estimate (e.g., (Kloeden & Platen, 1992, Thm. 10.2.2)) yields

$$\left( \mathbb{E}\|\boldsymbol{X}_1^{\mathrm{EM}} - \boldsymbol{X}_1\|^2 \right)^{1/2} \leq C_{\mathrm{EM}} \left( \int_0^1 (L_b(t)^2 + L_\sigma(t)^2)\, \mathrm{d}t \right)^{1/2} \Delta t^{1/2}.$$

Because $\sigma$ does not depend on $\boldsymbol{x}$, $L_\sigma(t) = 0$, which gives the stated bound. $\square$

**Theorem A.6** (Fisher spectrum and conditioning). *Let $\boldsymbol{F} = \mathbb{E}[\nabla_{\boldsymbol{\theta}} \ell \nabla_{\boldsymbol{\theta}} \ell^\top]$ be the Fisher (or generalized Gauss–Newton) matrix associated with the score matching loss. Assume the score network $S_{\boldsymbol{\theta}}(\cdot, t)$ has input Jacobian $\boldsymbol{J}_{\boldsymbol{\theta}}(\boldsymbol{x}, t) = \partial S_{\boldsymbol{\theta}}(\boldsymbol{x}, t)/\partial \boldsymbol{x}$ satisfying*

$$\|\boldsymbol{J}_{\boldsymbol{\theta}}(\boldsymbol{x}, t)\| \leq C_{\mathrm{net}}(t)\, \|\boldsymbol{x}\| \quad \forall \boldsymbol{x}, t.$$

*Then:*

    1. (**Spectral bound**) *The largest eigenvalue of $\boldsymbol{F}$ scales as*

$$\lambda_{\max}(\boldsymbol{F}) = \begin{cases} O(n), & \text{normalized Laplacian domain } \boldsymbol{S}, \\ O(n\Delta_{\max}^2), & \text{adjacency domain } \boldsymbol{A}. \end{cases}$$

    2. (**Condition number**) *If in addition $\lambda_{\min}(\boldsymbol{F}) \geq \gamma > 0$, then*

$$\kappa(\boldsymbol{F}) = \frac{\lambda_{\max}(\boldsymbol{F})}{\lambda_{\min}(\boldsymbol{F})} = \begin{cases} O(n/\gamma), & \boldsymbol{S}, \\ O(n\Delta_{\max}^2/\gamma), & \boldsymbol{A}. \end{cases}$$

*Proof.* Let $\ell(\boldsymbol{\theta}; \boldsymbol{X}_t, t)$ denote the score-matching loss at time $t$, with gradient

$$\nabla_{\boldsymbol{\theta}} \ell(\boldsymbol{\theta}; \boldsymbol{X}_t, t) = \boldsymbol{J}_{\boldsymbol{\theta}}(\boldsymbol{X}_t, t)^\top \big( S_{\boldsymbol{\theta}}(\boldsymbol{X}_t, t) - s(\boldsymbol{X}_t, t) \big),$$

where $s(\cdot, t)$ is the ground-truth score. **Step 1 (upper bound).** For any unit vector $\boldsymbol{u}$, the Rayleigh–Ritz principle gives

$$\boldsymbol{u}^\top \boldsymbol{F} \boldsymbol{u} = \mathbb{E}\big[ \langle \nabla_{\boldsymbol{\theta}} \ell, \boldsymbol{u} \rangle^2 \big] \leq \mathbb{E}\|\nabla_{\boldsymbol{\theta}} \ell\|^2.$$

Hence $\lambda_{\max}(\boldsymbol{F}) \leq \mathbb{E}\|\nabla_{\boldsymbol{\theta}} \ell\|^2$. By submultiplicativity,

$$\|\nabla_{\boldsymbol{\theta}} \ell\| \leq \|\boldsymbol{J}_{\boldsymbol{\theta}}(\boldsymbol{X}_t, t)\| \, \|S_{\boldsymbol{\theta}}(\boldsymbol{X}_t, t) - s(\boldsymbol{X}_t, t)\|.$$

Using the Jacobian bound, this yields

$$\|\nabla_{\boldsymbol{\theta}} \ell\|^2 \leq C_{\mathrm{net}}(t)^2 \, \|\boldsymbol{X}_t\|^2 \, \|S_{\boldsymbol{\theta}}(\boldsymbol{X}_t, t) - s(\boldsymbol{X}_t, t)\|^2.$$

Taking expectations and bounding the training error term by a finite constant $C_{\mathrm{err}} = \sup_t \mathbb{E}\|S_{\boldsymbol{\theta}}(\boldsymbol{X}_t, t) - s(\boldsymbol{X}_t, t)\|^2$ a.e, we obtain

$$\lambda_{\max}(\boldsymbol{F}) \leq C_{\mathrm{err}} \, \mathbb{E}[C_{\mathrm{net}}(t)^2 \, \|\boldsymbol{X}_t\|^2]. \tag{8}$$

**Step 2 (domain scaling of $\mathbb{E}\|\boldsymbol{X}_t\|^2$).** The forward corruption process is $\boldsymbol{X}_t = \sqrt{\bar{\alpha}_t} \boldsymbol{X}_0 + \sigma_t \varepsilon$, $\varepsilon \sim \mathcal{N}(\boldsymbol{0}, \boldsymbol{I}_n)$. Then

$$\mathbb{E}\|\boldsymbol{X}_t\|^2 = \bar{\alpha}_t \, \mathbb{E}\|\boldsymbol{X}_0\|^2 + n\sigma_t^2.$$

By Definition A.2, $\mathbb{E}\|\boldsymbol{X}_0\|^2 = O(n)$ in domain $\boldsymbol{S}$ and $O(n\Delta_{\max}^2)$ in domain $\boldsymbol{A}$. Thus the scaling of $\lambda_{\max}(\boldsymbol{F})$ in equation 8 matches the theorem. **Step 3 (condition number).** If $\lambda_{\min}(\boldsymbol{F}) \geq \gamma > 0$, then

$$\kappa(\boldsymbol{F}) = \frac{\lambda_{\max}(\boldsymbol{F})}{\lambda_{\min}(\boldsymbol{F})} = \begin{cases} O(n/\gamma), & \boldsymbol{S}, \\ O(n\Delta_{\max}^2/\gamma), & \boldsymbol{A}. \end{cases}$$

This completes the proof. $\qquad\square$

# B ADDITIONAL INFORMATION OF ESGD

## B.1 DEGREE MATRIX RECOVERY

We begin by analyzing the structure of the SNL $\boldsymbol{S} = -(\boldsymbol{D}')^{-1/2}\boldsymbol{A}(\boldsymbol{D}')^{-1/2}$. For an undirected, weighted graph with no self-loops:

$$\boldsymbol{S}_{i,j} = \begin{cases} 0 & \text{if } i = j \text{ (since } \boldsymbol{A}_{i,i} = 0 \text{ for no self-loops)} \\ -\frac{A_{i,j}}{\sqrt{d_i d_j}} & \text{if } i \neq j \text{ and } (i,j) \in \boldsymbol{E} \\ 0 & \text{if } i \neq j \text{ and } (i,j) \notin \boldsymbol{E} \end{cases} \tag{9}$$

where $d_i$ represents the unweighted degree of node $i$, which is simply the number of edges connected to node $i$ (regardless of their weights), $A_{i,j}$ is the weight of the edge between nodes $i$ and $j$, and $\boldsymbol{E}$ is the set of edges. For a weighted graph with unweighted degree matrix, when nodes $i$ and $j$ are adjacent:

$$S_{i,j} = -\frac{A_{i,j}}{\sqrt{d_i d_j}} \tag{10}$$

This means that for any edge $(i,j) \in \boldsymbol{E}$, the product of the degrees $d_i$ and $d_j$ is related to $\boldsymbol{S}$ and the edge weight $A_{i,j}$:

$$d_i d_j = \frac{A_{i,j}^2}{S_{i,j}^2} \tag{11}$$

For any node $i$ with at least two neighbors $j, k \in \mathcal{N}(i)$, we have:

$$\frac{d_j}{d_k} = \frac{S_{i,k}^2 \cdot A_{i,j}^2}{S_{i,j}^2 \cdot A_{i,k}^2} \tag{12}$$

Since the graph is connected, we can establish proportional relationships between all node degrees by traversing the graph. This gives us a system of equations that determines the degrees up to a constant factor. To resolve this remaining degree of freedom, we use the fact that the sum of all unweighted degrees equals twice the number of edges:

$$\sum_{i=1}^{n} d_i = 2|\mathrm{E}| \tag{13}$$

The number of edges $|\mathrm{E}|$ can be determined from the structure of $\boldsymbol{S}$ by counting the number of non-zero off-diagonal elements and dividing by 2. This yields a system of equations that uniquely determines the degree matrix $\boldsymbol{D}'$.

**Practical algorithm for estimating unweighted degree matrix:** In practical applications, the generated $\boldsymbol{S}$ may contain numerical errors or noise. Theoretically, elements corresponding to non-edges should be exactly zero, but in practice, they might appear as small non-zero values due to stochastic sampling process. Therefore, we introduce a thresholding parameter $\delta$ to distinguish between actual edges and numerical artifacts. The threshold parameter $\delta$ may need to be tuned based on the specific characteristics of the graph.

## B.2 ESGD MODEL ARCHITECTURE

ESGD (Efficient Spectral Graph Diffusion) is a spectral graph diffusion model based on symmetric normalized Laplacian matrices for graph generation tasks.

### B.2.1 CORE COMPONENTS

**SDE Framework:** The model employs Variance Preserving SDE (VPSDE):

$$d\mathbf{x} = -\frac{1}{2}\beta(t)\mathbf{x}dt + \sqrt{\beta(t)}d\mathbf{w} \tag{14}$$

$$\beta(t) = \beta_{min} + t(\beta_{max} - \beta_{min}) \tag{15}$$

where $\beta_{min} = 0.1$, $\beta_{max} = 20$, and $t \in [0, 1]$.

**Score Networks:** Two main networks predict scores for node features and adjacency matrices:

- *ScoreNetworkX*: Uses modified GCN layers with $\boldsymbol{S}$ convolution
- *ScoreNetworkA_eigen*: Operates in eigenvalue space with pooled node representations

**Modified GCN Layer:** Unlike traditional GCN, uses symmetric normalized Laplacian:

$$\boldsymbol{H}^{(l+1)} = \tanh\left(\boldsymbol{S}\boldsymbol{H}^{(l)}\boldsymbol{W}^{(l)}\right) \tag{16}$$

**Graph Multi-Head Attention:** Enhances representation with attention mechanism:

$$\boldsymbol{A}_{att} = \tanh\left(\frac{\boldsymbol{Q}\boldsymbol{K}^T}{\sqrt{d}}\right) \tag{17}$$

### B.2.2 LOSS FUNCTION

Score matching loss in both node and spectral domains:

$$\mathcal{L}(\boldsymbol{\theta}) = \frac{1}{2}\mathbb{E}_{t,\mathbf{x}_0,\boldsymbol{\epsilon}}\left[\left\|\mathbf{s}_{\boldsymbol{\theta}}(\mathbf{x}_t, t) + \frac{\boldsymbol{\epsilon}}{\sqrt{1-\alpha_t}}\right\|^2\right] \tag{18}$$

### B.2.3 KEY FEATURES

- Spectral domain diffusion for stability
- Support for both generic graphs and molecular generation
- Multiple $\beta(t)$ scheduling (linear, exponential, cosine)
- Computational complexity: $O(N^2 d + N d^2)$
- Datasets: Community-small, Grid, Enzymes, Ego-small, QM9, ZINC250k

# C EXPERIMENT DETAILS

In this section, we provide the detailed experimental settings. The hyperparameters of ESGD in this paper are provided in Table 7.

Table 7: Hyperparameters of ESGD used in the generic graph generation tasks and the molecule generation tasks. We provide the hyperparameters of the score-based models ($s_\theta$ and $s_\phi$), the diffusion processes (SDE for X and A), the SDE solver, and the training.

| | Hyperparameter | Ego-small | Community-small | Enzymes | Grid | Planar | SBM | Tree | QM9 | ZINC250k |
|---|---|---|---|---|---|---|---|---|---|---|
| $s_\theta$ | Number of GCN layers | 4 | 3 | 5 | 5 | 5 | 4 | 4 | 4 | 3 |
| | Hidden dimension | 32 | 32 | 32 | 32 | 32 | 32 | 32 | 16 | 16 |
| $s_\phi$ | Number of attention heads | 4 | 4 | 4 | 4 | 4 | 4 | 4 | 4 | 4 |
| | Number of initial channels | 2 | 2 | 2 | 2 | 2 | 2 | 2 | 2 | 2 |
| | Number of hidden channels | 8 | 8 | 8 | 8 | 8 | 8 | 8 | 8 | 8 |
| | Number of final channels | 4 | 4 | 4 | 4 | 4 | 4 | 4 | 4 | 4 |
| | Number of GCN layers | 5 | 5 | 7 | 7 | 7 | 6 | 7 | 6 | 6 |
| | Hidden dimension | 32 | 32 | 32 | 32 | 32 | 32 | 32 | 16 | 16 |
| SDE for X | Type | VP | VP | VP | VP | VP | VP | VP | VP | VP |
| | Number of sampling steps | 1000 | 1000 | 1000 | 1000 | 1000 | 1000 | 1000 | 1000 | 1000 |
| | $\beta_{min}$ | 0.1 | 0.1 | 0.1 | 0.1 | 0.1 | 0.1 | 0.1 | 0.1 | 0.1 |
| | $\beta_{max}$ | 1.0 | 1.0 | 1.0 | 1.0 | 1.0 | 1.0 | 1.0 | 10.0 | 4.0 |
| SDE for A | Type | VP | VP | VP | VP | VP | VP | VP | VP | VP |
| | Number of sampling steps | 1000 | 1000 | 1000 | 1000 | 1000 | 1000 | 1000 | 1000 | 1000 |
| | $\beta_{min}$ | 0.1 | 0.1 | 0.1 | 0.2 | 0.2 | 0.1 | 0.1 | 0.1 | 0.2 |
| | $\beta_{max}$ | 1.0 | 1.0 | 1.0 | 0.8 | 0.9 | 1.0 | 1.0 | 1.0 | 1.0 |
| Solver | Type | EM | EM + Langevin | EM + Langevin | EM + Langevin | EM + Langevin | EM + Langevin | EM + Langevin | EM + Langevin | EM + Langevin |
| | SNR | – | 0.05 | 0.2 | 0.1 | 0.10 | 0.15 | 0.10 | 0.2 | 0.2 |
| | Scale coefficient | – | 0.8 | 0.9 | 0.7 | 0.8 | 0.6 | 0.6 | 0.9 | 0.9 |
| Train | Optimizer | Adam | Adam | Adam | Adam | Adam | Adam | Adam | Adam | Adam |
| | Learning rate | $1 \times 10^{-2}$ | $1 \times 10^{-2}$ | $1 \times 10^{-2}$ | $1 \times 10^{-2}$ | $1 \times 10^{-2}$ | $1 \times 10^{-3}$ | $1 \times 10^{-2}$ | $5 \times 10^{-3}$ | $5 \times 10^{-3}$ |
| | Weight decay | $1 \times 10^{-4}$ | $1 \times 10^{-4}$ | $1 \times 10^{-4}$ | $1 \times 10^{-4}$ | $1 \times 10^{-4}$ | $1 \times 10^{-4}$ | $1 \times 10^{-4}$ | $1 \times 10^{-4}$ | $1 \times 10^{-4}$ |
| | Batch size | 128 | 128 | 64 | 8 | 64 | 32 | 128 | 1024 | 1024 |
| | Number of epochs | 5000 | 200 | 5000 | 5000 | 1000 | 3000 | 1000 | 300 | 500 |
| | EMA | – | – | 0.999 | 0.999 | 0.999 | 0.999 | 0.999 | – | – |

## C.1 DETAILS OF DATASETS

In this section, we provide key statistics of the datasets employed in the experiments, as shown in Table 8, for a better illustration of the experimental results. The statistics include the graph number in each dataset, the range of node numbers, the range of edge numbers for each node, the number of edge types, and the maximum eigenvalue.

Table 8: Statistics for the datasets in our experiments.

| | Name | Graph Number | Node range | Edge number of node | Number of edge types | Maximum eigenvalue |
|---|---|---|---|---|---|---|
| Generic | Ego-small | 200 | [4, 18] | [1, 16] | 1 | 9.036 |
| | Community-small | 100 | [12, 20] | [1, 9] | 1 | 6.6145 |
| | Enzymes | 587 | [10, 125] | [1, 9] | 1 | 5.3045 |
| | Grid | 100 | [100, 400] | [1, 4] | 1 | 3.9454 |
| | Planar | 200 | [64, 64] | [2, 12] | 1 | 6.1230 |
| | SBM | 200 | [44, 187] | [1, 23] | 1 | 14.1320 |
| | Tree | 200 | [64, 64] | [1, 8] | 1 | 3.0510 |
| Molecule | QM9 | 133,885 | [2, 9] | [1, 4] | 3 | 3.7063 |
| | ZINC250k | 249,455 | [6, 38] | [1, 4] | 3 | 3.5823 |
| Large | Cora | 1 | [2708, 2708] | [5429, 5429] | 1 | - |
| | Citeseer | 1 | [3312, 3312] | [4715, 4715] | 1 | - |
| | PubMed | 1 | [19717, 19717] | [44338, 44338] | 1 | - |

Note: For large citation networks, "Graph Number" is 1 since each dataset consists of a single giant graph. Node and edge ranges reduce to single values (the total counts).

## C.2 DETAILS OF EGO-SUBGRAPH DECOMPOSITION

To make training on large graphs feasible, we employ an ego-subgraph decomposition strategy implemented with NetworkX's `ego_graph` function. Given a center node and a radius $r$, an ego-subgraph contains the center and all nodes within $r$-hop distance, together with induced edges. We

apply size filters ($50 \leq |V| \leq 400$) to control computational complexity, remove self-loops, and relabel nodes to contiguous IDs. Datasets are split into training and test sets with an 80/20 ratio.

Table 9 reports the aggregated statistics of the constructed ego-subgraph datasets.

Table 9: Statistics of ego-subgraph datasets derived from large citation networks.

| Dataset | Num. subgraphs | Node range | Avg. nodes | Edge range | Avg. edges | Avg. degree |
|---------|----------------|------------|------------|------------|------------|-------------|
| Cora | 100 | 50–219 | 112.8 | 65–428 | 207.2 | 3.67 |
| Citeseer | 80 | 51–300 | 141.7 | 65–788 | 271.0 | 3.82 |
| PubMed | 100 | 50–282 | 112.5 | 60–1177 | 236.8 | 4.21 |

This decomposition provides three main benefits:

- **Efficiency**: smaller subgraphs reduce quadratic spectral costs and fit within GPU memory.
- **Structural fidelity**: local neighborhood motifs and degree/clustering statistics are preserved.
- **Generalization**: sampling multiple ego-subgraphs introduces data augmentation, mitigating overfitting to a single global graph.

### C.3 IMPLEMENTATION DETAILS FOR THE EXPERIMENTS ON GENERIC DATASETS

To evaluate the generated graphs, we employ the maximum mean discrepancy (MMD) to compare distributions of graph statistics between generated and test graphs. The evaluated statistics include degree, clustering coefficient, and occurrences of 4-node orbits. We compute the MMDs using the Gaussian Earth Mover's Distance (EMD) kernel on Ego-small, Community-small, Enzymes, and Grid following (Jo et al., 2022) and using the Gaussian Total Variation Distance (TV) kernel on Planar, SBM, and Tree following QIN et al. (2025).

As the setting from (Jo et al., 2022), we report the results of ESGD and GSDM on the Ego-small and Community-small datasets by 15 runs, 3 runs for 5 independently trained models, and on the Enzymes and Grid datasets by 3 runs. For GSDM, we use the hyperparameters given by the original paper and further search for the best performance if specific parameters do not exist. To get the best hyperparameters, we perform a grid search to choose the best signal-to-noise ratio (SNR) in $\{0.05, 0.1, 0.15, 0.2, 0.25, 0.3\}$ and the scale coefficient in the $\{0.1, 0.2, 0.3, 0.4, 0.5, 0.6, 0.7, 0.8, 0.9, 1.0\}$. We select the best MMD with the lowest average performance in Deg., Clus., and Orbit, respectively. Following (Jo et al., 2022), we quantize the value of each edge in the sampled adjacency matrix with the operator $1_{x>0.5}$ to get the 0-1 adjacency matrix. The specific hyperparameters are shown in Table 7.

### C.4 IMPLEMENTATION DETAILS FOR THE EXPERIMENTS ON MOLECULE DATASETS

We assess the quality of 10,000 generated graphs using multiple metrics. Frechet ChemNet Distance (FCD) leverages activations from ChemNet's penultimate layer to calculate the distance between test and generated graphs (Preuer et al., 2018). Neighborhood Subgraph Pairwise Distance Kernel (NSPDK) MMD measures the maximum mean discrepancy between test and generated graphs, accounting for both node and edge features (Costa & De Grave, 2010). Additionally, we report validity metrics: validity w/o correction and Validity represent the fractions of valid molecules without and with valency correction or edge resampling, respectively.

As the setting of (Jo et al., 2022), we report the results of ESGD and GSDM on QM9 and ZINC250k by 3 runs. We preprocess each molecule into a graph with the node features $X \in \{0, 1\}^{N \times F}$ and the adjacency matrix $A \in \{0, 1, 2, 3\}^{N \times N}$, where $N$ is the maximum number of atoms and $F$ is the number of atom types. We also use the grid search for the best SNR in $\{0.5, 1, 1.5, 2, 2.5, 3\}$ and the scale coefficient in $\{0.1, 0.2, 0.3, 0.4, 0.5, 0.6, 0.7, 0.8, 0.9, 1.0\}$. The specific hyperparameters are shown in Table 7. We select the hyperparameters for the best FCD value. We quantize the entries of the adjacency matrices to $\{0, 1, 2, 3\}$ by clipping the value $(-\infty, 0.5)$ to 0, $[0.5, 1.5)$ to 1, $[1.5, 2.5)$ to 2, and $[2.5, \infty)$ to 3 following (Jo et al., 2022).

## C.5 Computing resources

For all experiments, we use PyTorch to implement ESGD and train the score models on an NVIDIA RTX A4000 GPU with intel i7-14700K CPU.

# D  EXPERIMENTS MAKEUP AND MODIFICATION

Table 10: Generic graph generation on Community-small, Enzymes, Grid, and Ego-small. * The results were obtained by executing the published source code. Other results are taken from the published papers Luo et al. (2024); Wen et al. (2024); Jang et al. (2024); Eijkelboom et al. (2024). Hyphen (-) denotes that results are not provided and were not applicable due to memory issues. The best results are highlighted in **bold**, and the underline denotes the second best. We provide the standard deviations in Appendix E.1 due to page limit.

| | Community-small | | | | Enzymes | | | | Grid | | | | Ego-small | | | |
| | Synthetic, ($12 \leq V \leq 20$) | | | | Real, ($10 \leq V \leq 125$) | | | | Synthetic, ($100 \leq V \leq 400$) | | | | Real, ($4 \leq V \leq 18$) | | | |
| | Deg.↓ | Clus.↓ | Orbit↓ | Avg.↓ | Deg.↓ | Clus.↓ | Orbit↓ | Avg.↓ | Deg.↓ | Clus.↓ | Orbit↓ | Avg.↓ | Deg.↓ | Clus.↓ | Orbit↓ | Avg.↓ |
|---|---|---|---|---|---|---|---|---|---|---|---|---|---|---|---|---|
| DeepGMG Li et al. (2018) | 0.220 | 0.950 | 0.400 | 0.053 | - | - | - | - | - | - | - | - | 0.040 | 0.100 | 0.020 | 0.053 |
| GraphRNN You et al. (2018) | 0.080 | 0.120 | 0.040 | 0.080 | 0.017 | 0.062 | 0.046 | 0.042 | 0.064 | 0.043 | 0.021 | _0.043_ | 0.090 | 0.220 | 0.010 | 0.107 |
| GraphAF Shi et al. (2020) | 0.180 | 0.200 | 0.020 | 0.133 | 1.669 | 1.283 | 0.266 | 1.073 | - | - | - | - | 0.030 | 0.110 | 0.006 | 0.049 |
| GraphDF Luo et al. (2021) | 0.060 | 0.120 | 0.030 | 0.070 | 1.503 | 1.061 | 0.202 | 0.922 | - | - | - | - | 0.040 | 0.130 | 0.010 | 0.060 |
| GraphVAE | 0.350 | 0.980 | 0.540 | 0.623 | 1.369 | 0.629 | 0.191 | 0.730 | 1.619 | **0.000** | 0.919 | 0.846 | 0.130 | 0.170 | 0.050 | 0.117 |
| GNF Liu et al. (2019) | 0.200 | 0.200 | 0.110 | 0.170 | - | - | - | - | - | - | - | - | 0.030 | 0.100 | 0.006 | 0.045 |
| EDP-GNN Niu et al. (2020a) | 0.053 | 0.144 | 0.026 | 0.074 | 0.023 | 0.268 | 0.082 | 0.124 | 0.455 | 0.238 | 0.328 | 0.340 | 0.052 | 0.093 | 0.007 | 0.051 |
| WSGM Guth et al. (2022) | 0.039 | 0.084 | 0.009 | 0.044 | 0.034 | 0.097 | 0.013 | 0.048 | 0.083 | 0.006 | 0.065 | 0.051 | - | - | - | - |
| GDSS Jo et al. (2022) | 0.045 | 0.086 | 0.007 | 0.046 | 0.026 | 0.102 | _0.009_ | 0.046 | 0.111 | 0.005 | 0.070 | 0.062 | 0.021 | 0.024 | 0.007 | 0.017 |
| HGDM Wen et al. (2024) | _0.014_ | _0.050_ | 0.005 | 0.024 | 0.045 | _0.049_ | **0.003** | 0.032 | 0.137 | _0.004_ | 0.048 | 0.063 | _0.015_ | _0.023_ | _0.003_ | _0.014_ |
| GSDM* Luo et al. (2024) | 0.016 | 0.027 | 0.004 | 0.020 | 0.098 | 0.091 | 0.085 | 0.091 | _0.001_ | **0.000** | **0.000** | **0.000** | 0.027 | 0.034 | 0.004 | 0.023 |
| GEEL Jang et al. (2024) | - | - | - | - | **0.005** | **0.018** | _0.006_ | **0.010** | **0.000** | **0.000** | **0.000** | **0.000** | - | - | - | - |
| CatFlow Eijkelboom et al. (2024) | 0.018 | 0.086 | 0.007 | 0.037 | - | - | - | - | - | - | - | - | _0.013_ | 0.024 | 0.008 | 0.015 |
| GGSD* Minello et al. (2025) | 0.027 | 0.082 | 0.011 | 0.040 | - | - | - | - | - | - | - | - | - | - | - | - |
| ESGD (ours) | **0.005** | **0.006** | **0.000** | **0.004** | **0.005** | _0.026_ | **0.003** | _0.011_ | **0.000** | **0.000** | **0.000** | **0.000** | **0.005** | **0.021** | **0.002** | **0.009** |

Table 11: Generic graph generation on Planar, SBM, and Tree. Results are taken from the published papers Jang et al. (2024); QIN et al. (2025); Bergmeister et al. (2024b).

| | Planar | | | | SBM | | | | Tree | | | |
| | Synthetic, ($|V| = 64$) | | | | Synthetic, ($31 \leq V \leq 187$) | | | | Synthetic, ($|V| = 64$) | | | |
| | Deg.↓ | Clus.↓ | Orbit↓ | Spec.↓ | Deg.↓ | Clus.↓ | Orbit↓ | Spec.↓ | Deg.↓ | Clus.↓ | Orbit↓ | Spec.↓ |
|---|---|---|---|---|---|---|---|---|---|---|---|---|
| GraphRNN You et al. (2018) | 0.0049 | 0.2779 | 1.2543 | 0.0459 | 0.0055 | 0.0584 | 0.0785 | 0.0065 | - | - | - | - |
| GRAN Liao et al. (2019) | 0.0007 | 0.0426 | 0.0009 | 0.0075 | 0.0113 | 0.0553 | 0.0540 | 0.0054 | 0.1884 | _0.0080_ | 0.0199 | 0.2751 |
| SPECTRE Martinkus et al. (2022a) | 0.0005 | 0.0785 | 0.0012 | 0.0112 | 0.0015 | 0.0521 | 0.0412 | 0.0056 | - | - | - | - |
| DiGress Vignac et al. (2023a) | 0.0007 | 0.0780 | 0.0079 | 0.0098 | 0.0018 | 0.0485 | 0.0415 | 0.0045 | 0.2678 | 0.0428 | _0.0097_ | 0.0123 |
| EDGE Chen et al. (2023b) | 0.0761 | 0.3229 | 0.7737 | 0.0957 | 0.0279 | 0.1113 | 0.0854 | 0.0251 | 0.0211 | 0.1207 | 0.0374 | 0.0438 |
| GDSS Jo et al. (2022) | 0.2500 | 0.3930 | 0.5870 | - | 0.4960 | 0.4560 | 0.7170 | - | - | - | - | - |
| GEEL* Jang et al. (2024) | 0.0006 | 0.0458 | **0.0000** | 0.0070 | 0.0034 | 0.0621 | **0.0000** | 0.0049 | - | - | - | - |
| DisCo Xu et al. (2024) | _0.0002_ | 0.0403 | 0.0009 | - | _0.0006_ | 0.0266 | 0.0510 | - | - | - | - | - |
| Cometh Siraudin et al. (2025) | 0.0006 | 0.0434 | 0.0016 | _0.0049_ | 0.0020 | 0.0498 | _0.0383_ | **0.0024** | - | - | - | - |
| DeFoG QIN et al. (2025) | 0.0005 | 0.0501 | 0.0006 | 0.0072 | _0.0006_ | 0.0517 | 0.0556 | 0.0054 | _0.0002_ | **0.0000** | **0.0000** | 0.0108 |
| Local PPGN (one-shot) Bergmeister et al. (2024b) | 0.0003 | _0.0245_ | 0.0006 | 0.0104 | 0.0141 | 0.0528 | 0.0809 | 0.0071 | 0.0004 | **0.0000** | **0.0000** | **0.0080** |
| Local PPGN Bergmeister et al. (2024b) | 0.0005 | 0.0626 | 0.0017 | 0.0075 | 0.0119 | 0.0517 | 0.0669 | 0.0067 | **0.0001** | **0.0000** | **0.0000** | 0.0117 |
| GGSD* Minello et al. (2025) | 0.0024 | 0.0807 | 0.0048 | **0.0048** | 0.0041 | 0.0431 | 0.0730 | 0.0090 | - | - | - | - |
| ESGD (ours) | **0.0001** | **0.0228** | _0.0002_ | 0.0057 | **0.0005** | **0.0027** | 0.0462 | _0.0039_ | **0.0000** | 0.0001 | **0.0000** | _0.0081_ |

Table 12: Sampling results with statistical analysis across multiple random seeds (mean ± std over 5 runs).

| Dataset | Degree↓ | Cluster↓ | Orbit↓ | Spectral↓ | Validity↑ | Uniqueness↑ | Novelty↑ |
|---|---|---|---|---|---|---|---|
| *ESGD (Ours)* | | | | | | | |
| ENZYMES | 0.0049±0.0008 | 0.0263±0.0041 | 0.0027±0.0005 | 0.0213±0.0032 | 0.932±0.024 | 0.853±0.031 | 1.000±0.000 |
| Community | 0.0052±0.0011 | 0.0064±0.0015 | 0.0003±0.0001 | 0.0500±0.0078 | 1.000±0.000 | 0.550±0.045 | 0.500±0.052 |
| Ego | 0.0045±0.0009 | 0.0208±0.0033 | 0.0024±0.0004 | 0.0729±0.0112 | 0.900±0.031 | 0.500±0.041 | 0.444±0.058 |
| Grid | 0.0000±0.0000 | 0.0000±0.0000 | 0.0000±0.0000 | 0.0257±0.0039 | 1.000±0.000 | 0.700±0.037 | 0.650±0.043 |
| Planar | 0.0001±0.0000 | 0.0228±0.0036 | 0.0002±0.0001 | 0.0057±0.0009 | 1.000±0.000 | 0.925±0.028 | 1.000±0.000 |
| SBM | 0.0005±0.0001 | 0.0027±0.0009 | 0.0462±0.0098 | 0.0039±0.0010 | 0.975±0.018 | 0.872±0.029 | 1.000±0.000 |
| Tree | 0.0000±0.0000 | 0.0001±0.0000 | 0.0000±0.0000 | 0.0081±0.0012 | 1.000±0.000 | 0.875±0.029 | 0.775±0.041 |

Table 13: Random seeds used for experiments across different datasets.

| Dataset | Random Seeds |
|---|---|
| Community-small | 34941, 82137, 86966, 5683, 39812 |
| Ego-small | 13022, 15400, 28451, 4360, 19692 |
| ENZYMES | 81925, 49667, 45730, 63059, 91579 |
| Grid | 7517, 5740, 79457, 74714, 12309 |
| Planar | 36429, 90321, 61220, 30920, 602 |
| SBM | 53237, 41582, 67839, 28914, 95273 |
| Tree | 87562, 42318, 65491, 13847, 79205 |

Table 14: 95% Confidence intervals for key metrics.

| Dataset | Degree↓ | Cluster↓ | Spectral↓ | Validity↑ |
|---|---|---|---|---|
| ENZYMES | [0.0038, 0.0060] | [0.0210, 0.0316] | [0.0168, 0.0258] | [0.898, 0.966] |
| Community | [0.0038, 0.0066] | [0.0044, 0.0084] | [0.0398, 0.0602] | [1.000, 1.000] |
| Ego | [0.0033, 0.0057] | [0.0164, 0.0252] | [0.0585, 0.0873] | [0.856, 0.944] |
| Grid | [0.0000, 0.0000] | [0.0000, 0.0000] | [0.0206, 0.0308] | [1.000, 1.000] |
| Planar | [0.0001, 0.0001] | [0.0182, 0.0274] | [0.0045, 0.0069] | [1.000, 1.000] |
| SBM | [0.0002, 0.0005] | [0.0027, 0.0069] | [0.0039, 0.0066] | [0.950, 1.000] |
| Tree | [0.0000, 0.0000] | [0.0001, 0.0001] | [0.0064, 0.0098] | [1.000, 1.000] |

# E ADDITIONAL EXPERIMENTAL RESULTS

In this section, we provide additional experimental results.

## E.1 GENERIC GRAPH GENERATION

We report the standard deviation of the generation results of Table 1 in Table 15 and Table 16. We provide sampling acceleration of ESGD on the Community-small and Ego-small datasets in Table 17.

Table 15: Generation results of ESGD on Ego-small and Community-small. $^*$ denotes that the results are obtained by running open-source codes. The results of GDSS and HGDM are taken from (Luo et al., 2024; Wen et al., 2024; Eijkelboom et al., 2024). The best results are highlighted in **bold** (lower is better), and the underline denotes the second best. We report the MMD distance between the test datasets and the generated graphs with the standard deviation.

| | Ego-small | | | Community-small | | |
|---|---|---|---|---|---|---|
| | Deg.↓ | Clus.↓ | Orbit↓ | Deg.↓ | Clus.↓ | Orbit↓ |
| GDSS (Jo et al., 2022) | 0.021±0.008 | 0.024±0.007 | 0.007±0.005 | 0.045±0.028 | 0.086±0.022 | 0.007±0.004 |
| HGDM (Wen et al., 2024) | 0.015±0.005 | 0.023±0.006 | 0.003±0.005 | 0.017±0.029 | 0.050±0.018 | 0.005±0.003 |
| GSDM$^*$ (Luo et al., 2024) | 0.027±0.000 | 0.034±0.007 | 0.004±0.001 | 0.016±0.018 | 0.027±0.026 | 0.004±0.005 |
| CatFlow Eijkelboom et al. (2024) | 0.013±0.007 | 0.024±0.009 | **0.001**±0.005 | 0.018±0.012 | 0.086±0.021 | 0.007±0.005 |
| ESGD (Ours) | **0.009**±0.003 | **0.022**±0.002 | **0.001**±0.000 | **0.007**±0.003 | **0.010**±0.004 | **0.001**±0.000 |

## E.2 MOLECULE GENERATION

We additionally report the validity, uniqueness, and novelty of the generated molecules aside from the results in Table 4 to comprehensively illustrate the performance of molecule generation. Validity is the fraction of the generated molecules that do not violate the chemical valency rule. Uniqueness is the fraction of the valid molecules that are unique. Novelty is the fraction of the valid molecules that are not in the training set. Moreover, the standard deviation of each metric is also provided in this section. The results of molecule generation are shown in Table 18 and Table 19.

Table 16: Generation results of ESGD on Enzymes and Grid. * denotes that the results are obtained by running open-source codes. The results of GDSS and HGDM are taken from (Luo et al., 2024; Wen et al., 2024; Eijkelboom et al., 2024). The best results are highlighted in **bold** (lower is better), and the underline denotes the second best. We report the MMD distance between the test datasets and the generated graphs with the standard deviation.

| | Enzymes | | | Grid | | |
|---|---|---|---|---|---|---|
| | Deg.↓ | Clus.↓ | Orbit↓ | Deg.↓ | Clus.↓ | Orbit↓ |
| GDSS (Jo et al., 2022) | 0.026±0.008 | 0.102±0.010 | 0.009±0.005 | 0.111±0.012 | 0.004±0.000 | 0.070±0.044 |
| HGDM (Wen et al., 2024) | 0.045±0.008 | **0.049**±0.011 | **0.003**±0.001 | 0.137±0.019 | 0.004±0.000 | 0.070±0.044 |
| GSDM* (Luo et al., 2024) | 0.098±0.010 | 0.091±0.003 | 0.085±0.010 | 0.001±0.000 | **0.000**±0.000 | **0.000**±0.000 |
| CatFlow Eijkelboom et al. (2024) | 0.013±0.012 | 0.062±0.011 | 0.008±0.007 | 0.115±0.010 | 0.004±0.002 | 0.075±0.071 |
| ESGD (ours) | **0.007**±0.001 | 0.064±0.002 | 0.009±0.001 | **0.000**±0.000 | **0.000**±0.000 | **0.000**±0.000 |

Table 17: Sampling efficiency of ESGD by 1 run on Community-small and Ego-small.

| Dataset | Steps | Deg.↓ | Clus.↓ | Orbit↓ | Time (s)↓ |
|---|---|---|---|---|---|
| Community-small | 1000 | 0.011 | 0.015 | 0.001 | 1.51 |
| | 500 | 0.011 | 0.015 | 0.001 | 0.94 |
| | 250 | 0.011 | 0.015 | 0.001 | 0.58 |
| | 200 | 0.058 | 0.106 | 0.012 | 0.48 |
| Ego-small | 1000 | 0.012 | 0.019 | 0.001 | 1.13 |
| | 800 | 0.014 | 0.014 | 0.001 | 0.9 |
| | 750 | 0.015 | 0.014 | 0.001 | 0.86 |
| | 700 | 0.018 | 0.029 | 0.003 | 0.82 |

As shown in Table 18, the generated molecules of ESGD have lower novelty and comparable uniqueness. As discussed in (Wen et al., 2024), high novelty does not necessarily represent good generation quality due to the property of the QM9 dataset, such as the generated molecules of GraphDF and GraphEBM. In other words, the models that can generate molecules with high novelty fail to capture adequate properties of the dataset. Also as discussed in Vignac & Frossard (2022) QM9 is an exhaustive enumeration of the small molecules that satisfy a given set of constrains, generating molecules outside this set is not necessarily a good sign that the network has correctly captured the data distribution.

Table 18: Generation results on QM9. * denotes that the results are obtained by running open-source codes. Other results of the baselines are taken from the published papers Luo et al. (2024); Wen et al. (2024); Jang et al. (2024); Eijkelboom et al. (2024). The best results are highlighted in **bold**, and the underline denotes the second best.

| Method | Val. w/o (%)↑ | NSPDK MMD↓ | FCD↓ | Validity↑ | Uniqueness↑ | Novelty↑ |
|---|---|---|---|---|---|---|
| GraphAF (Shi et al., 2020) | 67 | 0.020±0.003 | 5.268±0.403 | **100.00** | 94.51 | 88.83 |
| GraphDF (Luo et al., 2021) | 82.67 | 0.063±0.001 | 10.816±0.020 | **100.00** | 97.62 | **98.10** |
| MoFlow (Zang & Wang, 2020) | 91.36±1.23 | 0.017±0.003 | 4.467±0.595 | **100.00**±0.00 | 98.65±0.57 | 94.72±0.77 |
| EDP-GNN (Niu et al., 2020a) | 47.52±3.60 | 0.005±0.001 | 2.680±0.221 | **100.00**±0.00 | 99.25±0.05 | 86.58±1.85 |
| GDSS (Jo et al., 2022) | 95.79±1.93 | 0.003±0.000 | 2.813±0.278 | **100.00**±0.00 | 98.02±0.63 | 82.55±3.11 |
| HGDM (Wen et al., 2024) | 98.04±1.27 | 0.002±0.000 | 2.13±0.254 | **100.00**±0.00 | 97.27±0.71 | 69.63±2.75 |
| GSDM* (Luo et al., 2024) | 99.81±0.08 | 0.009±0.000 | 3.191±0.014 | **100.00**±0.00 | 94.7±0.15 | 68.5±0.47 |
| CatFlow Eijkelboom et al. (2024) | 99.81 ± 0.03 | - | 0.441±0.023 | **100.00** ± 0.00 | **99.95** ± 0.02 | - |
| GEEL Jang et al. (2024) | **100.0** | **0.0002** | **0.089** | **100.00** | 96.08 | 22.30 |
| ESGD (ours) | 99.20±0.02 | 0.002±0.000 | 1.425±0.009 | **100.00**±0.00 | 96.61±0.16 | 60.64±0.00 |

We present acceleration results for ESGD sampling with 1,000-step training on both QM9 and ZINC250k datasets in Table 20. For QM9, ESGD maintains comparable accuracy when using 600 sampling steps. Although validity scores without correction decrease at lower step counts, both NSPDK and FCD metrics remain robust even when using as few as 500 sampling steps. This demonstrates that ESGD maintains excellent sampling efficiency when applied to molecule datasets.

Table 19: Generation results on ZINC250k. $^*$ denotes that the results are obtained by running open-source codes. Other results of the baselines are taken from the published papers Luo et al. (2024); Wen et al. (2024); Jang et al. (2024); Eijkelboom et al. (2024); QIN et al. (2025). The best results are highlighted in **bold**, and the underline denotes the second best.

| Method | Val. w/o (%)↑ | NSPDK MMD↓ | FCD↓ | Validity↑ | Uniqueness↑ | Novelty↑ |
|---|---|---|---|---|---|---|
| GraphAF (Shi et al., 2020) | 68 | 0.044±0.006 | 16.289±0.482 | **100.00** | 99.10 | **100.00** |
| GraphDF (Luo et al., 2021) | 89.03 | 0.176±0.001 | 34.202±0.160 | **100.00** | 99.16 | **100.00** |
| MoFlow (Zang & Wang, 2020) | 63.11±5.17 | 0.046±0.002 | 20.931±0.184 | **100.00**±0.00 | 99.99±0.01 | **100.00**±0.00 |
| EDP-GNN (Niu et al., 2020a) | 82.97±2.73 | 0.049±0.006 | 16.737±1.300 | **100.00**±0.00 | 99.79±0.08 | **100.00**±0.00 |
| GDSS (Jo et al., 2022) | 95.90±1.01 | 0.019±0.001 | 16.621±1.213 | **100.00**±0.00 | 99.67±0.14 | **100.00**±0.00 |
| HGDM (Wen et al., 2024) | 93.51±0.87 | 0.016±0.001 | 17.69±1.146 | **100.00**±0.00 | 99.82±0.18 | **100.00**±0.00 |
| GSDM$^*$ (Luo et al., 2024) | 93.0±0.04 | 0.016±0.000 | 12.07±0.062 | **100.00**±0.00 | 99.97±0.09 | **100.00**±0.00 |
| CatFlow Eijkelboom et al. (2024) | 99.21 ± 0.04 | - | 13.211 ± 0.012 | **100.00** ± 0.00 | **100.00** ± 0.00 | - |
| GEEL Jang et al. (2024) | **99.31** | 0.0068 | **0.401** | **100.00** | 99.97 | 99.89 |
| DeFoG QIN et al. (2025) | 99.22 ± 0.08 | **0.0008** ± 0.0001 | 1.425 ± 0.0001 | **100.00** ± 0.00 | 99.99 ± 0.01 | - |
| ESGD (ours) | 98.29±0.58 | 0.010±0.000 | 8.80±0.132 | **100.00**±0.00 | 99.76±0.12 | **100.00**±0.00 |

Table 20: Sampling efficiency of ESGD by 1 run on QM9 and ZINC250k.

| Dataset | Steps | Val. w/o (%)↑ | FCD↓ | NSPDK MMD↓ | Time (s)↓ |
|---|---|---|---|---|---|
| QM9 | 1000 | 99.23 | 1.421 | 0.002 | 14.3 |
| | 800 | 99.35 | 1.427 | 0.002 | 11.2 |
| | 600 | 99.22 | 1.485 | 0.002 | 8.1 |
| | 500 | 99.08 | 1.595 | 0.003 | 5.4 |
| ZINC250k | 1000 | 98.81 | 8.856 | 0.011 | 71.6 |
| | 800 | 97.34 | 8.623 | 0.011 | 60.2 |
| | 500 | 95.94 | 8.813 | 0.010 | 37.5 |
| | 400 | 95.44 | 8.989 | 0.010 | 30.6 |

# F  VISUALIZATION

## F.1  GENERIC GRAPH GENERATION

We visualize a randomly selected subset of samples from the training datasets and the generated graph set in Figures 4-10.

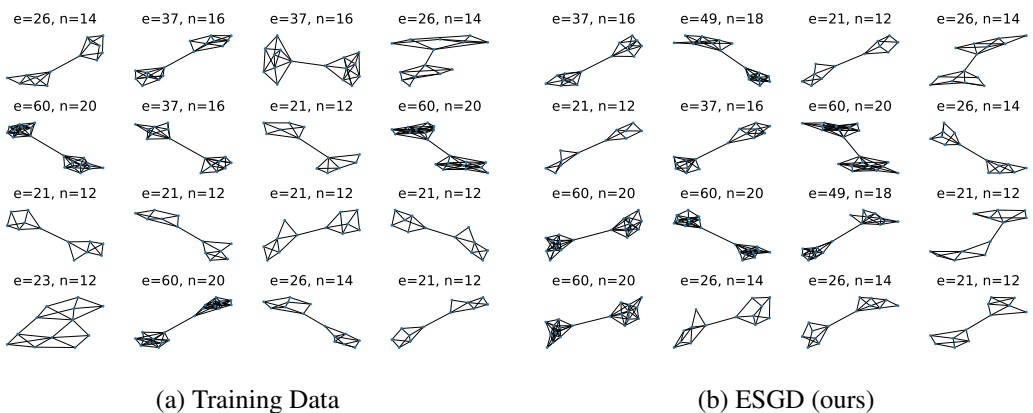

(a) Training Data                    (b) ESGD (ours)

Figure 4: Visualization of the graphs from the Community-small dataset and the generated graphs of ESGD.

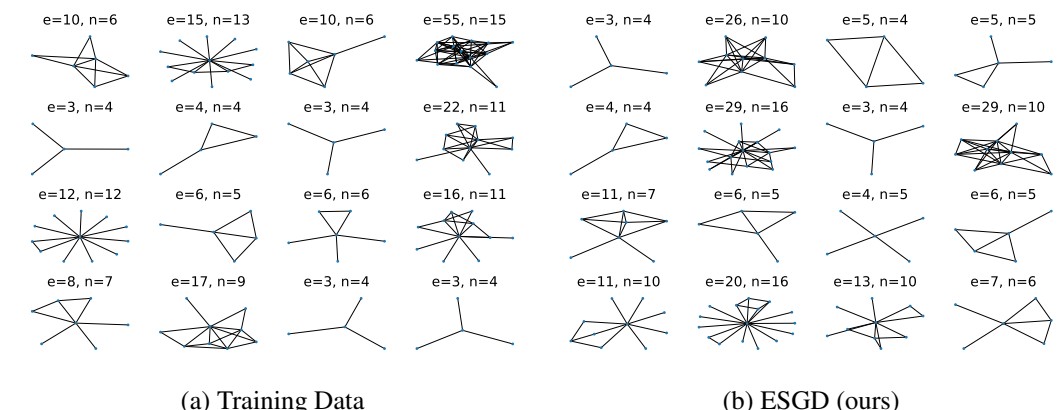

(a) Training Data                    (b) ESGD (ours)

Figure 5: Visualization of the graphs from the Ego-small dataset and the generated graphs of ESGD.

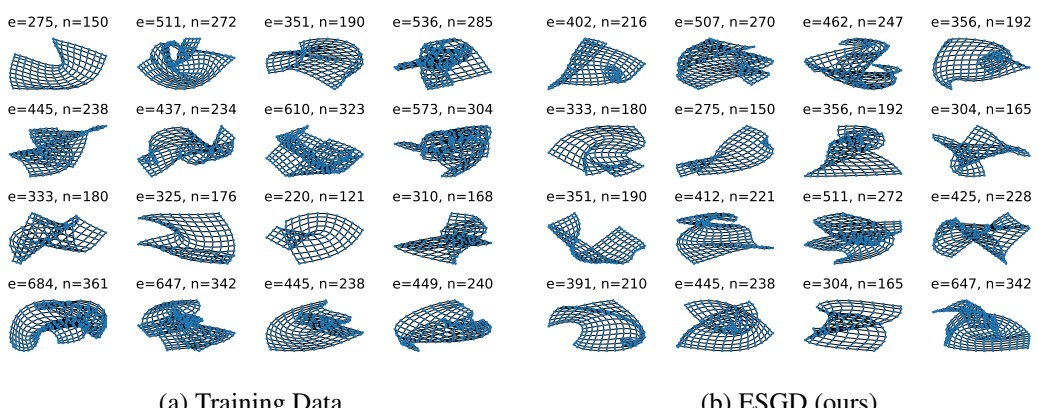

(a) Training Data                    (b) ESGD (ours)

Figure 6: Visualization of the graphs from the Grid dataset and the generated graphs of ESGD.

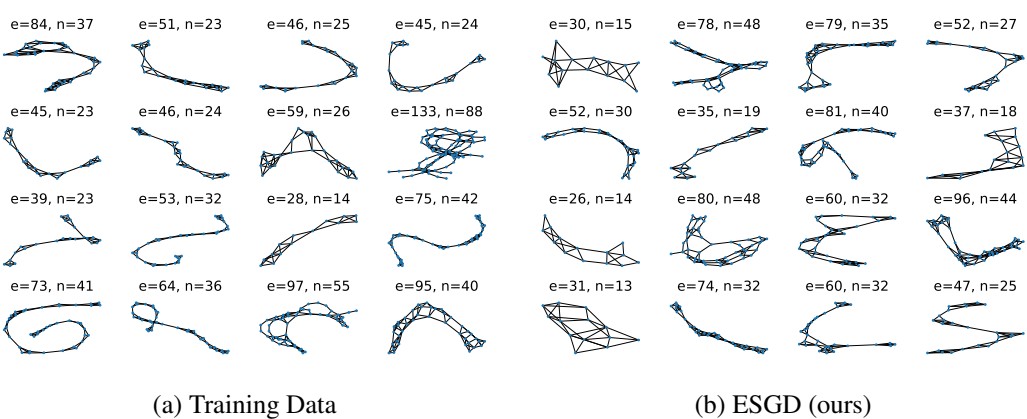

(a) Training Data                    (b) ESGD (ours)

Figure 7: Visualization of the graphs from the Enzymes dataset and the generated graphs of ESGD.

### F.2    MOLECULE GRAPH GENERATION

We visualize a randomly selected subset of the generated graph set in Figures 11-12.

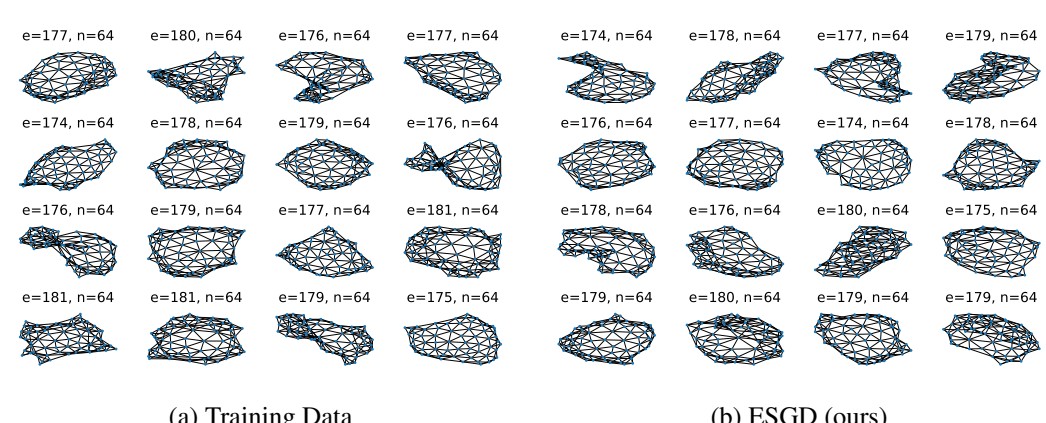

(a) Training Data               (b) ESGD (ours)

Figure 8: Visualization of the graphs from the Planar dataset and the generated graphs of ESGD.

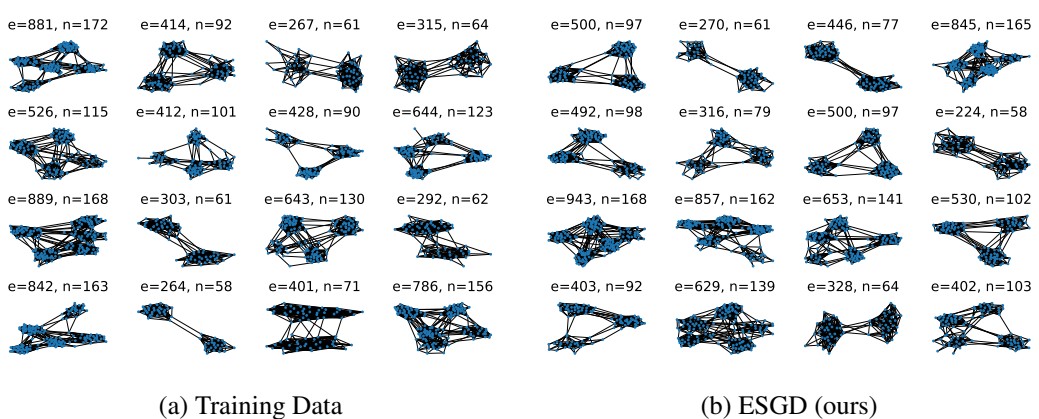

(a) Training Data               (b) ESGD (ours)

Figure 9: Visualization of the graphs from the SBM dataset and the generated graphs of ESGD.

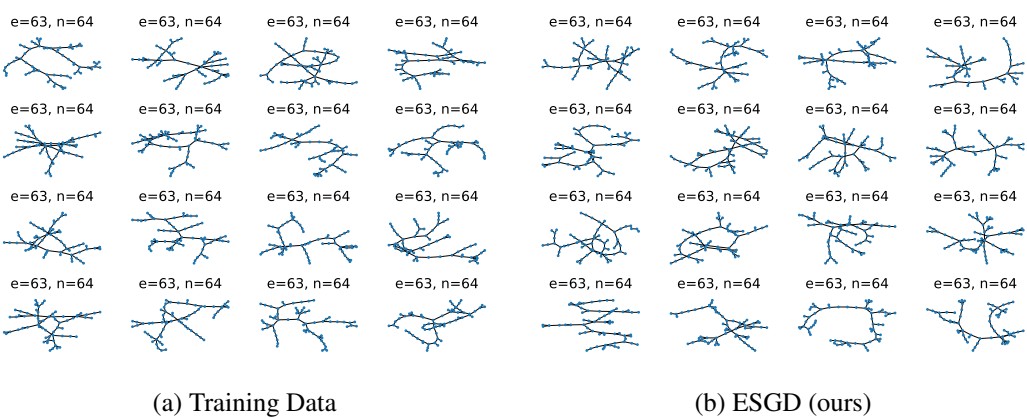

(a) Training Data               (b) ESGD (ours)

Figure 10: Visualization of the graphs from the Tree dataset and the generated graphs of ESGD.

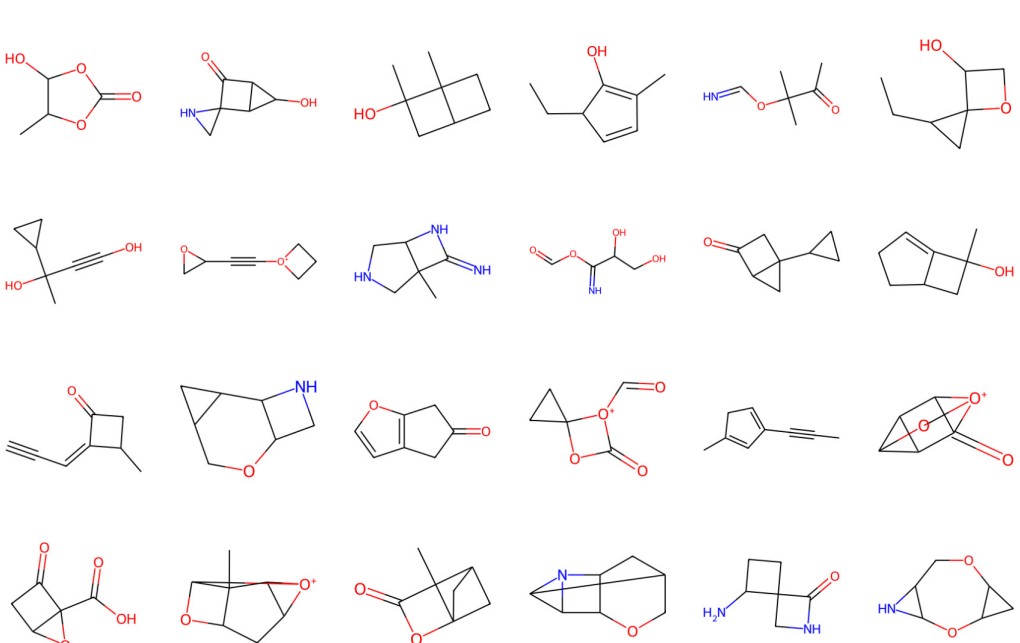

Figure 11: Visualization of the random samples generated by ESGD trained on QM9.

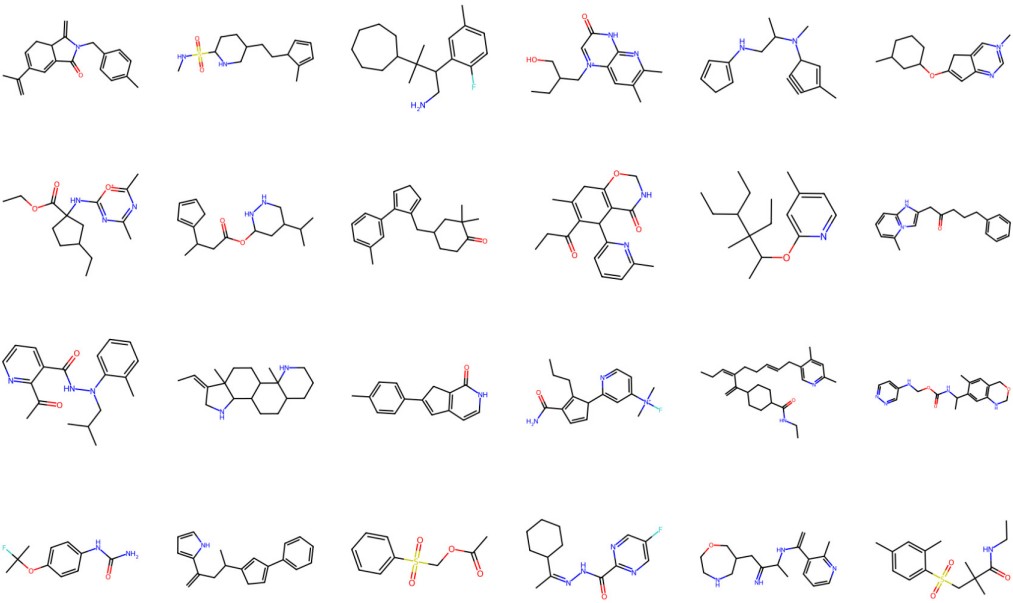

Figure 12: Visualization of the random samples generated by ESGD trained on ZINC250k.

