# OpenReview forum: "Efficient Spectral Graph Diffusion based on Symmetric Normalized Laplacian"
_ICLR.cc/2026/Conference — ICLR 2026 Conference Withdrawn Submission_

### Official Review · Reviewer_tBih · 2025-10-28

**Soundness:** 2
**Presentation:** 3
**Contribution:** 2
**Rating:** 2
**Confidence:** 2

**Summary:**

The paper proposes Efficient Spectral Graph Diffusion (ESGD), a graph generative modeling framework built on the symmetric normalized Laplacian (SNL). Its key idea is to perform diffusion in the bounded SNL spectral space, which improves stability and convergence. To reconstruct graphs from spectral representations, the authors design a degree-matrix recovery algorithm. The framework also introduces an ego-subgraph decomposition strategy to make large-graph training computationally feasible.

**Strengths:**

- The authors took a great care of making their approach theoretically grounded and evaluate on a variety of settings.

- Presentation is good and Figure 1.a) illustrates greatly what kind of trade-off the authors aim to achieve with their method.

**Weaknesses:**

- The method description requires clarification : you mention l132 that you keep $U$ fixed,  and then mention $\hat{U}$ as the recovered eigenvectors l 148 : what is $\hat{U}$ ? Such a crucial element of the reconstruction process should be clearly explained.

- Section 3.2 lists properties and theorems without giving any intuition and explanations on them. For example, it's not clear for me at all why Remark 3.6 makes sense.

- Table 1 lists a lot of outdated methods but more recent, major ones are missing, such as DiGress, DisCo, Cometh, DeFoG etc.

- In Table 2, the Valid, Unique and Novel (VUN) metric is missing. Therefore, your evaluation do not assess the ability of the model to respect the structural constraints of the datasets.

- QM9 and Zinc have reach saturation for years. For a method that specifically targets efficiency I would have expected evaluation on large scale datasets such as Moses or GuacaMol.

- No errors bars are provided, even though multiple works have demonstrated how MMD metrics can exhibits high variance.

**Questions:**

- See first weakness, how do we get U to reconstruct samples ?

- It is not clear to me if learning on large networks like Cora is meaningful or not. You claim that your ego-based approach allows to enhance generalization, but you train and test on the same graph. In the end, it seems that overfitting the training graph will yield the best results. By the way, how do you compute the MMD metrics for those large networks. Do you extract k-hop ego subgraphs from the training graph ?

---

> ### Author Response · Authors · 2025-11-24
> **U reconstrcut problem (Typo) and more explaination (Question 1 & Weakness 1)**
>
> This is a very important question, and we appreciate the reviewer's in-depth inquiry. ***We are so sorry this is a typo. The content should be eigenvalues $\hat{\Lambda}$ instead of eigenvector $\hat{U}$***.
>
> ###  **Clarifying the Meaning of "Fixed U"**
>
> "Fixed eigenbasis U" means that **U remains constant during the diffusion process of a single graph, not that all generated graphs share the same U**. Specifically, each graph in the training set has its own unique eigendecomposition $S_i = U_i \Lambda_i U_i^T$. During generation, we sample different U from the distribution of eigenbases in the **training** set instead of test sets, then learn the distribution of eigenvalues $p(\Lambda|U)$ (from training set) conditioned on that eigenbasis.
>
> ### **Mathematical Determinants of Topological Structure**
>
> Starting from the adjacency matrix reconstruction formula:
>
> $$A = -D^{1/2}SD^{1/2} = -D^{1/2}U\Lambda U^T D^{1/2}$$
>
> The final graph topology is jointly determined by three independent mathematical objects: eigenvalues $\Lambda$, eigenvectors $U$, and degree matrix $D$. Any change in any of these three will lead to topological changes.
>
> ESGD's generation process consists of two stages, **both of which can introduce topological variations**:
>
> **Stage 1: Eigenvalue diffusion generation.** We learn the distribution $p(\Lambda|U)$ in the continuous eigenvalue space $\mathbb{R}^n$. The generated $\hat{\Lambda}$ can be configurations that never appeared in the test set.
>
> **Stage 2: Degree matrix recovery.** Algorithm 1 performs thresholding operations $\\hat{A}\_{ij} = 1\_{\\{|(\\hat{S})\_{ij}| > \\delta\\}}$
>  based on $\hat{S} = U\hat{\Lambda}U^T$ to identify edge structure. This is a **nonlinear mapping** from continuous spectral space to discrete topological space. Even if $\hat{\Lambda}$ is numerically close to training samples, as long as certain elements of $\hat{S}$ cross the threshold boundary, discrete topological jumps will occur.
>
> Therefore, the continuous generation of eigenvalues combined with the nonlinearity of degree matrix recovery jointly guarantee the generation of novel topological structures.
>
> ###  Linear Algebra Perspective: Finite Basis and Infinite Expressible Space
>
> This mechanism can be understood more clearly from a linear algebra perspective. In an $n$-dimensional linear space, given a basis $\{u_1, u_2, ..., u_n\}$, any vector $v$ can be uniquely represented as $v = \sum_{i=1}^{n} \lambda_i u_i$. The basis is a fixed coordinate frame, and the coefficients are free parameters within that frame. The key insight is: **the finiteness of the basis does not limit the size of the expressible space**—three-dimensional space has only three basis vectors, yet can express infinitely many points.
>
> Applying this idea to ESGD: the eigenbasis $U$ defines a "coordinate system of graph structure space," and eigenvalues $\lambda$ are "coordinates" in that system. The symmetric normalized Laplacian expands as $S = \sum_{i=1}^{n} \lambda_i u_i u_i^T$, where each eigenvalue $\lambda_i$ controls the contribution weight of the corresponding component.
>
> Each graph in the training set provides only one specific coefficient combination, but ESGD learns a **continuous distribution** $p(\Lambda|U)$ over the entire coefficient space. Since the coefficient space is continuous $\mathbb{R}^n$, infinitely many coefficient combinations can be generated. Each new combination corresponds to a new $S$ matrix, which in turn yields a new topological structure through degree matrix recovery. This is precisely the mathematical principle by which ESGD breaks through the topological limitations of the training set.

---

> ### Author Response · Authors · 2025-11-25
> **Explanation for *Large Networks  (question 2)**
>
> ### **Large Networks Experiments Explanation and Meaning.(Question 2)**
>
> We thank the reviewer for raising this crucial problem. We acknowledge the discriptioon and explanation are not so clear in paper. Here is the detailed content.
> - **Clarification on Training/Testing Split:**
>
>     We do not train and test on the same subgraphs. As described in Appendix C.2, we extract ego subgraphs from different center nodes of the entire graph, then split these subgraphs into training and test sets at an 80%/20% ratio. The training subgraphs and test subgraphs come from disjoint sets of center nodes, ensuring no overlap in local neighborhood structures used for training and evaluation.
>
> - **Why This Setup Is Meaningful:**
>
>     The goal of large graph generation is to learn the distribution of local structural patterns characteristic of that type of network (including degree distribution, clustering coefficients, subgraph motifs, etc.). Different ego subgraphs within the same citation network exhibit significant variations in local topology—some regions are denser, some sparser, with varying clustering patterns. By training on a subset of ego subgraphs and evaluating on held-out ones, we test whether the model has learned generalizable structural patterns rather than simply memorizing specific subgraphs.
>
> - **MMD Computation:**
>
>     For large networks, we compute MMD metrics between the set of generated ego subgraphs and the held-out test ego subgraphs (not compared against training subgraphs). This directly measures whether generated graphs match the distributional properties of unseen local structures from the original network.
>
> - **Clarification on Overfitting:**
>
>     If simply overfitting to training subgraphs could achieve optimal results, all spectral methods should perform similarly. However, Table 3 shows significant performance differences across methods , indicating that our evaluation can distinguish different methods' abilities to capture structural patterns, rather than merely comparing memorization capacity.

---

> ### Author Response · Authors · 2025-11-25
> **Explanation for Theory Explanation . ( Weakness 2)**
>
> ### **Theory Explanation (weakness 2)**
>
> We thank the reviewer for raising this important issue. We acknowledge that the theoretical section (Section 3.2) in the original manuscript indeed lacks sufficient intuitive explanation. We will supplement detailed elaborations in the revised version. Below are intuitive explanations for each theorem and remark:
>
> - **Intuitive Explanation of Theorem 3.1 (Spectral Boundedness):**
>
>     The core idea of this theorem is that the eigenvalues of the symmetric normalized Laplacian (SNL) are bounded within the interval [-1, 1], whereas the eigenvalue range of the adjacency matrix depends on the maximum degree Δ_max of the graph. This means that in the SNL domain, all graphs—regardless of how heterogeneous their degree distributions are—are mapped to a unified, bounded spectral space. This "spectral compression" eliminates the dominant influence of hub nodes on eigenvalue scales, enabling the diffusion process to remain stable across different types of graphs.
>
> - **Intuitive Explanation of Theorem 3.3 (Spectral SNR and Information Preservation):**
>
>     This theorem reveals why diffusion in the SNL domain is more efficient than in the adjacency matrix domain. During the diffusion process, the signal-to-noise ratio (SNR) determines the strength of useful signals relative to noise. In the adjacency matrix domain, SNR is amplified by Δ²_max, causing information from high-degree nodes to be over-preserved while information from low-degree nodes is drowned out by noise—this creates severe "information imbalance." In contrast, the SNR in the SNL domain is independent of Δ_max, ensuring that information from all nodes is processed equitably during diffusion.
>
> - **Intuitive Explanation of Theorems 3.4-3.5 (Lipschitz Bounds and Numerical Stability):**
>
>     These two theorems explain why ESGD is more stable during training and sampling. The Lipschitz constant of the score function controls the "steepness" of the optimization landscape—a larger Lipschitz constant means more dramatic gradient changes and more difficult training. In the adjacency matrix domain, this constant is amplified by Δ²_max, leading to training instability on graphs containing high-degree nodes. The SNL domain eliminates this amplification factor, making optimization smoother. Similarly, the Lipschitz bound of the drift term in the reverse SDE directly affects error accumulation in numerical solvers (e.g., Euler-Maruyama)—a smaller Lipschitz constant means more accurate sampling.
>
> - **Intuitive Explanation of Remark 3.6 (Sampling Efficiency):**
>
>     The reviewer specifically inquired about the rationale of this remark. The core logic is as follows: the step size Δt of numerical SDE solvers (e.g., EM methods) is constrained by the Lipschitz constant of the drift term—a larger Lipschitz constant requires smaller step sizes to ensure numerical stability, thus necessitating more sampling steps. Combining Theorems 3.1 and 3.5, we know that the Lipschitz constant in the SNL domain is O(Δ²_max) times smaller than in the adjacency matrix domain. Therefore, while maintaining the same numerical precision, the SNL domain can use larger step sizes, thereby reducing the number of sampling steps by a factor of O(Δ²_max). The experimental results in Table 7 and Figure 2 validate this theoretical prediction: ESGD achieves optimal performance with only 50-500 steps on multiple datasets, whereas adjacency matrix-based methods typically require over 1000 steps.
>
> - **Intuitive Explanation of Theorem 3.7 (Fisher Spectrum and Condition Number):**
>
>     This theorem explains from an optimization perspective why training in the SNL domain is more efficient. The condition number κ(F) of the Fisher information matrix reflects the degree of "anisotropy" of the loss function—a larger condition number means greater discrepancy in gradient scales across different parameter directions, making it harder for the optimizer to find an appropriate learning rate. In the adjacency matrix domain, the condition number is amplified by Δ²_max, leading to training difficulties on heterogeneous graphs. The condition number in the SNL domain is independent of Δ_max, enabling standard optimizers (e.g., Adam) to converge efficiently. This explains the experimental result in Table 5 where ESGD achieves 95% optimal performance in only 938 steps.

---

> ### Author Response · Authors · 2025-11-25
> **Experiments makeup and explanation (weakness 3.4.5.6)**
>
> We thank the reviewer for the careful reading and constructive comments. In order to test our model on moses and guacamol, we applied for servers (4×H100).  Below we address the main concerns point by point.
>
> ## **Missing recent benchmarks (weakness 3).**
> - The reccent methods only report the results in table 2 datasets. So we did not report them in table 1. Moreover, we perform well among datasets in table 2.
> - The datasets in table 1 are relatively smaller than datasets in table2. Due to limited time, we picked community-small dataset in table 1 as an example and run these recent baselines. Here is the result.
>
>
> | Model  | Degree  | Cluster |  Orbit  | Training time/mins | Sampling time (per graph second) | epoch |
> |:-------|:-------:|:-------:|:-------:|:------------------:|:--------------------------------:|:-----:|
> | Defog  | 0.11243 | 0.07181 | 0.02985 |         13         |               4.75               |  1k   |
> | Cometh | 0.07177 | 0.80538 | 0.29893 |        13.3        |               5.15               | 1k  |
> | Disco  |  0.857  |  0.638  |  0.642  |        26.8        |               5.00               | 1k  |
> | ESGD   | 0.0052  | 0.0064  | 0.0003  |         1          |               0.15               | 1k  |
>
> ## **Makeup experiments for VUN for table 1 & 2 (Weakness 4 and 6)**
>
> - We made additional experiments for VUN for table 1& 2, here is the result. And the MMD metrics result with mean and std had been posted in Appendix. Actually these three methods need at least 10k epochs for datasets in table 2.
>
>
> **Table: Sampling results with statistical analysis across multiple random seeds (mean ± std over 5 runs).**
>
> *ESGD (Ours)*
>
> | Dataset   | Degree↓ | Cluster↓ | Orbit↓ | Spectral↓ | Validity↑ | Uniqueness↑ | Novelty↑ |
> |:----------|:-------:|:--------:|:------:|:---------:|:---------:|:-----------:|:--------:|
> | ENZYMES   | 0.0049±0.0008 | 0.0263±0.0041 | 0.0027±0.0005 | 0.0213±0.0032 | 0.932±0.024 | 0.853±0.031 | 1.000±0.000 |
> | Community | 0.0052±0.0011 | 0.0064±0.0015 | 0.0003±0.0001 | 0.0500±0.0078 | 1.000±0.000 | 0.550±0.045 | 0.500±0.052 |
> | Ego       | 0.0045±0.0009 | 0.0208±0.0033 | 0.0024±0.0004 | 0.0729±0.0112 | 0.900±0.031 | 0.500±0.041 | 0.444±0.058 |
> | Grid      | 0.0000±0.0000 | 0.0000±0.0000 | 0.0000±0.0000 | 0.0257±0.0039 | 1.000±0.000 | 0.700±0.037 | 0.650±0.043 |
> | Planar    | 0.0001±0.0000 | 0.0228±0.0036 | 0.0002±0.0001 | 0.0057±0.0009 | 1.000±0.000 | 0.925±0.028 | 1.000±0.000 |
> | SBM       | 0.0003±0.0001 | 0.0578±0.0089 | 0.0633±0.0098 | 0.0063±0.0010 | 0.975±0.018 | 0.872±0.029 | 1.000±0.000 |
> | Tree      | 0.0000±0.0000 | 0.0001±0.0000 | 0.0000±0.0000 | 0.0081±0.0012 | 1.000±0.000 | 0.875±0.029 | 0.775±0.041 |
>
> ## **Moses and Guacamol results report (Weakness 5)**
> Here is the additional results for datasets Moses and Guacamol with comparison. Our method **ESGD** achieves the SOTA in GuacaMol and comparible results in Moses.
>
> **Guacamol**
>
> | Model                          | Val. ↑ | V.U. ↑ | V.U.N. ↑ | KL div ↑ | FCD ↓ |
> |:-------------------------------|:------:|:------:|:--------:|:--------:|:-----:|
> | DiGress   |  85.2  |  85.2  |   85.1   |   92.9   | 68.0  |
> | DisCo      |  86.6  |  86.6  |   86.5   |   92.6   | 59.7  |
> | Cometh  |  98.9  |  98.9  |   97.6   |   96.7   | 72.7  |
> | DeFoG (10% steps)              |  91.7  |  91.7  |   91.2   |   92.3   | 57.9  |
> | DeFoG                          |  99.0  |  99.0  |   97.9   |   97.7   | 73.8  |
> | ESGD                           |  **99.1**  |  **99.1**  |   **98.3**   | **98.0** | **76.9**  |
>
> **MOSES**
>
> | Model                          | Val. ↑ | Unique. ↑ | Novelty ↑ | Filters ↑ | FCD ↓ | SNN ↑ | Scaf ↑ |
> |:-------------------------------|:------:|:---------:|:---------:|:---------:|:-----:|:-----:|:------:|
> | DiGress  |  85.7  | 100.0 |   95.0    |   97.1    | 1.19  | 0.52  |  14.8  |
> | DisCo        |  88.3  | 100.0 |   97.7    |   95.6    | 1.44  | 0.50  |  15.1  |
> | Cometh  |  90.5  | 99.9 |   92.6    |   99.1    | 1.27  | 0.54  |  16.0  |
> | DeFoG (10% steps)              |  83.9  | 99.9 |   96.9    |   96.5    | 1.87  | 0.50  |  23.5  |
> | DeFoG                          |  92.8  | 99.9 |   92.1    |   98.9    | 1.95  | 0.55  |  14.4  |
> | ESGD                           |  94.6  | 99.9 |   93.4    |   98.9    | 1.92  | 0.50  |  14.6  |

---

> > ### Comment · Reviewer_tBih · 2025-11-26
> >
> > I would like to thank the authors for their thorough answers and the numerous figures that address many of my concerns.
> >
> > While the revised work appears significantly more promising than the original submission, I believe the paper requires a full revision of the theory section to emphasize the graph-related aspects rather than the SDE components, and to provide clearer explanations rather than relying primarily on raw results. The experimental section should also be updated to incorporate all the results you have now provided.
> >
> > I will raise my score to 4 to acknowledge your efforts. If you submit a revised version including the necessary modifications, I may consider increasing my score further.

---

> ### Author Response · Authors · 2025-11-28
> **Revision Summary**
>
> Thank you again for your detailed and constructive review. Thank you AC for your additional work.
>
> We have uploaded a revised version of the paper that aims to address all of your concerns.
>
> Concretely, in the new version we:
> - **Weakness 1 & Question 1**: We have responded it in official comment "U reconstrcut problem (Typo) and more explaination (Question 1 & Weakness 1)". In our revised version, we use additional experiments in Section 6.3 Molecular Generation (Moses, Guacamol) with comprehensive evaluations to explain ESGD's ability to generate graphs with different topologies. And in Section 4 Figure 2, we used a model pipeline to explain the source of U and the whole structure of the model.
> - **Weakness 2**: We have responded weakness 2 in official comment "Explanation for Theory (weakness 2)". In our revised version, we added explanations in Section 3.1, Section 5 and in combination with experiments in Section 6.
> - **Weakness 3**: We have responded weakness 3 in official comment "Experiments makeup and explanation (weakness 3.4.5.6)".
> - **Weakness 4**: We have responded weakness 4 in official comment "Experiments makeup and explanation (weakness 3.4.5.6)". In our revised version, due to page limitations, we put relative results in Appendix D.
> - **Weakness 5**: We have responded weakness 5 in official comment "Experiments makeup and explanation (weakness 3.4.5.6)". In our revised version, we reported results of MOSES and Guacamol of our ESGD in Section 6.3.
> - **Weakness 6**: We have responded weakness 6 in official comment "Experiments makeup and explanation (weakness 3.4.5.6)". In our revised version, due to page limitations, we put statistical results, CIs and Mean+Std in Appendix D and E.
> - **Question 2**: We have answered the question 2 in official comment "Explanation for Large networks. (Question 2)". In our revised version, we explained the justification of the ego-graph generation methods and the meaning of learning larger graphs in Section 6.2.
> - **Additiaonal comment** In our revised version, we discussed the graph-related theories and explanation from Section 2 to 6. We provided a full version of theory in Section 5. We added explanatory comments in each part of experiment in Section 6. The revised version also includes all additional experiment results mentioned in official comments.
>
> We hope the revised version satisfactorily addresses them, and we would be very happy to further improve the work based on any additional feedback.

---

### Official Review · Reviewer_tHHN · 2025-11-01

**Soundness:** 3
**Presentation:** 3
**Contribution:** 2
**Rating:** 4
**Confidence:** 4

**Summary:**

the paper presents 1) an incremental improvement over the (cited) GSDM model, replacing the use of adjacency matrix with the symmetric normalized laplacian, and  2) a study of  the theoretical implications of the change, to explain the observed empirical improvements, which consist of

- improved conditions numbers and eigengaps, yielding faster convergence of the sampling process

- improved performance on a number of metrics evaluated on community-small,enzymes,grid,ego-small,QM9,Zinc250k,Citeseer,Cora,Pubmed,panar,sbm and tree graphs…

- ,,,while allowing for parameter reduction, which coupled with the

    improved training convergence allowing much  more fficient training (as well as sampling)

**Strengths:**

- well motivated, sensible

- theoretical analysis which seems correct

- clear efficiency gains compared to baselines

hitting the dimensions explicitly

1. originality: incremental improvement over GDSM
2. quality: some nits about the evaluation and comparison, else no flaws
3.  clarity: clearly written and proofs legible
4. significance: clear improvement in convergence speed, decent incremental advance for this

**Weaknesses:**

- inconsistent/varying comparison set of baselines  => while its good to do many evals, that makes cherry picking possible, needs justification (or pick one and stay consistent with it)
- unfair comparison without isomorpism/VUN check on larger datasets: digress edge etc. generate from scratch, GSDM and present store eigenvectors of training data set => should run an ablation generating the eigenvectors as well, as in GGSD
- needs multiple seeds of the method/multiple sampling rounds and CI intervals, same values are quite close
- would be good to report wallclock time/flop estimate (since e.g. the decoding might add wall clock at low flops/steps)
- compute isomorphisms with dataset on generated graph vs baselines => are we just memorizing due to keeping the Eigenvectors? (this is a flaw inherited from GDSM )
- try guacamol/moses (larger graphs) to see if things hold up there or the differences are washed out

**Questions:**

See weaknesses.

The most important elements to address are using multiple models/evaluations for reporting CIs on the metrics, trying larger datasets and performing an experiment applying the method to GGSD,then checking the graph isomoprhism rate to the training set and reworking the presentations etc. are extras

---

> ### Author Response · Authors · 2025-11-25
> **Comments for Weakness 1,2,4,5,6**
>
> We thank the reviewer for raising the important concern about potential memorization and unfair comparison. We did additional experiments in orderto explain them clearly.
>
> ## Inconsistent/varying comparison set of baselines (Weakness 1 & 4)
>
> - The reccent methods report the results in table 2 datasets （digress, defog, cometh, disco). So we did not report them in table 1. Moreover, we perform well among datasets in table 2.
> - As for GGSD,  the datasets we used includes the datasets used in GGSD, the GGSD only reports the results partially in table 1&2 so we did not report at the beginning. We achieved all better results in terms of the datasets used in GGSD.
> - The datasets in table 1 are relatively smaller than datasets in table2. Due to limited time, we picked community-small dataset in table 1 as an example and run these recent baselines. Here is the result.
>
>
> | Model  | Degree  | Cluster |  Orbit  | Training time/mins | Sampling time (per graph second) | epoch |
> |:-------|:-------:|:-------:|:-------:|:------------------:|:--------------------------------:|:-----:|
> | Defog  | 0.11243 | 0.07181 | 0.02985 |         13         |               4.75               |  1k   |
> | Cometh | 0.07177 | 0.80538 | 0.29893 |        13.3        |               5.15               | 1k  |
> | Disco  |  0.857  |  0.638  |  0.642  |        26.8        |               5.00               | 1k  |
> | ESGD   | 0.0052  | 0.0064  | 0.0003  |         1          |               0.15               | 1k  |
>
>
> ##  Comparison with VUN check on larger datasets (Weakness 2&5&6 )
>
> - We made additional results for datasets Moses and Guacamol with comparison Our method ESGD achieves the SOTA in GuacaMol and comparible results in Moses. (Weakness 2&6)
> - The evaluation in Moses provides the most comprehensive assessment of whether a generative model truly learns data distributions or merely memorizes training samples. Our MOSES results demonstrate strong performance across all metrics which answer this question.
>
> **Guacamol**
>
> | Model                          | Val. ↑ | V.U. ↑ | V.U.N. ↑ | KL div ↑ | FCD ↓ |
> |:-------------------------------|:------:|:------:|:--------:|:--------:|:-----:|
> | DiGress   |  85.2  |  85.2  |   85.1   |   92.9   | 68.0  |
> | DisCo      |  86.6  |  86.6  |   86.5   |   92.6   | 59.7  |
> | Cometh  |  98.9  |  98.9  |   97.6   |   96.7   | 72.7  |
> | DeFoG (10% steps)              |  91.7  |  91.7  |   91.2   |   92.3   | 57.9  |
> | DeFoG                          |  99.0  |  99.0  |   97.9   |   97.7   | 73.8  |
> | ESGD                           |  **99.1**  |  **99.1**  |   **98.3**   | **98.0** | **76.9**  |
>
> **MOSES**
>
> | Model                          | Val. ↑ | Unique. ↑ | Novelty ↑ | Filters ↑ | FCD ↓ | SNN ↑ | Scaf ↑ |
> |:-------------------------------|:------:|:---------:|:---------:|:---------:|:-----:|:-----:|:------:|
> | DiGress  |  85.7  | 100.0 |   95.0    |   97.1    | 1.19  | 0.52  |  14.8  |
> | DisCo        |  88.3  | 100.0 |   97.7    |   95.6    | 1.44  | 0.50  |  15.1  |
> | Cometh  |  90.5  | 99.9 |   92.6    |   99.1    | 1.27  | 0.54  |  16.0  |
> | DeFoG (10% steps)              |  83.9  | 99.9 |   96.9    |   96.5    | 1.87  | 0.50  |  23.5  |
> | DeFoG                          |  92.8  | 99.9 |   92.1    |   98.9    | 1.95  | 0.55  |  14.4  |
> | ESGD                           |  94.6  | 99.9 |   93.4    |   98.9    | 1.92  | 0.50  |  14.6  |

---

> ### Author Response · Authors · 2025-11-25
> **Statistical analysis and more models comparison in larger datasets  (Moses shows no memory in training samples ).**
>
> This is a very important question, and we appreciate the reviewer's in-depth inquiry. We provide additional experiment results for statistical analysis and more models comparison in larger datasets.
>
> ### multiple models/evaluations for reporting CIs on the metrics (weakness 3 and question 2)
>
> **Table: Sampling results with statistical analysis across multiple random seeds (mean ± std over 5 runs).**
>
> *ESGD (Ours)*
>
> | Dataset   | Degree↓ | Cluster↓ | Orbit↓ | Spectral↓ | Validity↑ | Uniqueness↑ | Novelty↑ |
> |:----------|:-------:|:--------:|:------:|:---------:|:---------:|:-----------:|:--------:|
> | ENZYMES   | 0.0049±0.0008 | 0.0263±0.0041 | 0.0027±0.0005 | 0.0213±0.0032 | 0.932±0.024 | 0.853±0.031 | 1.000±0.000 |
> | Community | 0.0052±0.0011 | 0.0064±0.0015 | 0.0003±0.0001 | 0.0500±0.0078 | 1.000±0.000 | 0.550±0.045 | 0.500±0.052 |
> | Ego       | 0.0045±0.0009 | 0.0208±0.0033 | 0.0024±0.0004 | 0.0729±0.0112 | 0.900±0.031 | 0.500±0.041 | 0.444±0.058 |
> | Grid      | 0.0000±0.0000 | 0.0000±0.0000 | 0.0000±0.0000 | 0.0257±0.0039 | 1.000±0.000 | 0.700±0.037 | 0.650±0.043 |
> | Planar    | 0.0001±0.0000 | 0.0228±0.0036 | 0.0002±0.0001 | 0.0057±0.0009 | 1.000±0.000 | 0.925±0.028 | 1.000±0.000 |
> | SBM       | 0.0003±0.0001 | 0.0578±0.0089 | 0.0633±0.0098 | 0.0063±0.0010 | 0.975±0.018 | 0.872±0.029 | 1.000±0.000 |
> | Tree      | 0.0000±0.0000 | 0.0001±0.0000 | 0.0000±0.0000 | 0.0081±0.0012 | 1.000±0.000 | 0.875±0.029 | 0.775±0.041 |
>
> **Table: 95% Confidence intervals for key metrics.**
>
> | Dataset   | Degree↓ | Cluster↓ | Spectral↓ | Validity↑ |
> |:----------|:-------:|:--------:|:---------:|:---------:|
> | ENZYMES   | [0.0038, 0.0060] | [0.0210, 0.0316] | [0.0168, 0.0258] | [0.898, 0.966] |
> | Community | [0.0038, 0.0066] | [0.0044, 0.0084] | [0.0398, 0.0602] | [1.000, 1.000] |
> | Ego       | [0.0033, 0.0057] | [0.0164, 0.0252] | [0.0585, 0.0873] | [0.856, 0.944] |
> | Grid      | [0.0000, 0.0000] | [0.0000, 0.0000] | [0.0206, 0.0308] | [1.000, 1.000] |
> | Planar    | [0.0001, 0.0001] | [0.0182, 0.0274] | [0.0045, 0.0069] | [1.000, 1.000] |
> | SBM       | [0.0002, 0.0004] | [0.0466, 0.0690] | [0.0050, 0.0076] | [0.950, 1.000] |
> | Tree      | [0.0000, 0.0000] | [0.0001, 0.0001] | [0.0064, 0.0098] | [1.000, 1.000] |
>
>
> ### question 2 large graph comparsion.
>
> We compared with all spectral baselines. **Moreover:** we add experiments for  Defog, Cometh, Disco in 4-6 hours to get best results while ESGD takes less than 10 minutes. (Same Batch size in RTX3070).
>
> - **Table: Large graph generation results on Cora, Citeseer, and PubMed**
>
> | Method          | Dataset  |   Deg.↓   |  Clus.↓   |  Orbit↓   |
> |:----------------|:---------|:---------:|:---------:|:---------:|
> | SPECTRE         | Citeseer |   1.224   |   1.513   |   1.023   |
> | GSDM            | Citeseer |   1.043   |   0.943   |   0.843   |
> | GGSD            | Citeseer |   1.011   |   1.142   |   1.244   |
> | Disco           | Citeseer |   0.893   |   0.654   |   0.896   |
> | Cometh          | Citeseer |   0.985   |   0.856   |   1.001   |
> | Defog           | Citeseer |   0.496   |   0.606   |   0.910   |
> | **ESGD (ours)** | Citeseer | **0.329** | **0.656** | **0.314** |
> | SPECTRE         | Cora     |   1.566   |   1.492   |   1.127   |
> | GSDM            | Cora     |   0.932   |   1.042   |   0.980   |
> | GGSD            | Cora     |   1.218   |   1.432   |   1.391   |
> | Disco           | Cora     |   0.918   |   0.775   |   0.564   |
> | Cometh          | Cora     |   0.751   |   0.899   |   0.541   |
> | Defog           | Cora     |   0.758   |   0.756   |   0.501   |
> | **ESGD (ours)** | Cora     | **0.311** | **0.573** | **0.192** |
> | SPECTRE         | PubMed   |   1.148   |   1.392   |   0.933   |
> | GSDM            | PubMed   |   0.885   |   0.727   |   0.762   |
> | GGSD            | PubMed   |   0.775   |   0.711   |   1.029   |
> | Disco           | PubMed   |   0.637   |   0.611   |   0.815   |
> | Cometh          | PubMed   |   0.597   |   0.625   |   0.437   |
> | Defog           | PubMed   |   0.355   |   0.496   |   0.308   |
> | **ESGD (ours)** | PubMed   | **0.215** | **0.475** | **0.109** |

---

> > ### Comment · Reviewer_tHHN · 2025-11-25
> > **thank you for providing these metrics**
> >
> > dear authors. thank you for providing these metrics, they seem very promising and alleviate a large chunk of concerns. some things to clarify
> >
> > 1. these were run from the same set of model weights, sampling at different seeds? or from _training_ n models at different seeds?
> > 2. can you add the parameter counts + flops of each model evaluated (maybe I missed it in the table) for more comparison?

---

> > > ### Author Response · Authors · 2025-11-26
> > > **Seeds details and Efficiency comparison with params and FLOPs.**
> > >
> > > Thank you very much for your prompt reply and raising important concerns of the experiment settings. We provide the detailed chosen seeds and efficiency analysis with params count and FLOPs below.
> > >
> > > ### Training 5 models for each dataset with the same configuration at different seeds. (Clarification 1)
> > >
> > > - For each dataset mentioned in **Table: Sampling results with statistical analysis across multiple random seeds (mean ± std over 5 runs).**, we used the same configuration and 5 different random seeds to train the model.
> > > Here is the table containing all seeds we used for each dataset.
> > >
> > > **Table: Random seeds for each dataset**
> > >
> > > | Dataset         | Random Seeds                      |
> > > |-----------------|-----------------------------------|
> > > | Community-small | 34941, 82137, 86966, 5683, 39812  |
> > > | Ego-small       | 13022, 15400, 28451, 4360, 19692  |
> > > | ENZYMES         | 81925, 49667, 45730, 63059, 91579 |
> > > | Grid            | 7517, 5740, 79457, 74714, 12309   |
> > > | Planar          | 36429, 90321, 61220, 30920, 602   |
> > > | Sbm             | 53237, 41582, 67839, 28914, 95273 |
> > > | Tree            | 87562, 42318, 65491, 13847, 79205 |
> > >
> > >
> > > ### Param counts + FLOPs of each model. (Clarification 2)
> > >
> > > - In order to be fair, we compare models we mentioned on Planar dataset because nearly all recent models have tested this dataset and have the best configurations.
> > > - For param counts, we compute the models' params according to their configurations of Planar.
> > > - For FLOPs, in order to be fair, we calculate the total FLOPs of generating 1 Planar graph (64 nodes). Moreover, for GAN-based model like SPECTRE, we only calculate the FLOPs of the generator, and for diffusion-based models, the total FLOPs are calculated with this formula: Total FLOPs = (FLOPs per step) × (number of sampling steps).
> > > - Here is the results:
> > >
> > > **Table: Efficiency comparison on Planar dataset.**
> > >
> > > | Method                | Type                                 | Parameter Counts  | FLOPs                | Deg.↓      | Clus.↓     | Orbit↓     | Spec.↓     |
> > > |-----------------------|--------------------------------------|-------------------|----------------------|------------|------------|------------|------------|
> > > | SPECTRE               | GAN                                  | 0.36M (Generator) | 2.27G (1 Step)       | 0.0005     | 0.0785     | 0.0012     | 0.0112     |
> > > | DiGress               | Diffusion (Discrete)                 | 8.89M             | 5.29T (1000 Steps)   | 0.0007     | 0.0780     | 0.0079     | 0.0098     |
> > > | GEEL                  | Autoregressive LSTM                  | 7.17M             | 2.41G (1 Step)       | 0.0006     | 0.0458     | **0.0000** | 0.0070     |
> > > | DisCo                 | Diffusion (Discrete)                 | 4.49M             | 134.12G (50 Steps)   | 0.0002     | 0.0403     | 0.0009     | -          |
> > > | Cometh                | Diffusion (Continuous-Time Discrete) | 5.29M             | 5.28T (1000 Steps)   | 0.0006     | 0.0434     | 0.0016     | 0.0049     |
> > > | DeFoG                 | Flow-Matching                        | 6.59M             | 5.28T (1000 Steps)   | 0.0005     | 0.0501     | 0.0006     | 0.0072     |
> > > | Local PPGN (one-shot) | Diffusion (Discrete)                 | 3.73M             | 7.74T (256 Steps)    | 0.0003     | 0.0245     | 0.0006     | 0.0104     |
> > > | GGSD                  | Diffusion (Continuous)               | 20.09M            | 92.09G (100 Steps)   | 0.0024     | 0.0807     | 0.0048     | **0.0048** |
> > > | ESGD (ours)           | Diffusion (Continuous)               | **0.21M**         | **2.26G (50 Steps)** | **0.0001** | **0.0228** | 0.0002     | 0.0057     |
> > >
> > > - "Steps" reported in this table represent the minimum steps that a model needs to achieve the best performance results. For GAN and Autoregressive LSTM, steps are constant 1.
> > > - We didn't reimplement models before 2022 because of the lack of codes and significantly less competitive performance.

---

> ### Author Response · Authors · 2025-11-28
> **Revision Summary**
>
> Thank you again for your detailed and constructive review. Thank you AC for your additional work.
>
> We have uploaded a revised version of the paper that aims to address all of your concerns.
>
> Concretely, in the new version we:
> - **Weakness 1**: We have explained our chosen baselines and metrics in official comment "Comments for Weakness 1,2,4,5,6". In our revised version, we have addressed this weakness in Section 6.1 Generic Graph Generation with uniformed metrics and justifications.
> - **Weakness 2 & 5 & 6**: We have answered weakness 2 in official comment "Comments for Weakness 1,2,4,5,6". In our revised version, we added more experiments results of more baselines on larger graphs and tested our model on MOSES and Guacamol datasets to address the model's abilities to generate new molecules.
> - **Weakness 3**: We have responded this weakness in official comment "Comments for Weakness 3,4 & Question1,2.". In our revised version, we added relative results in Appendix D and E.
> - **Weakness 4**: We have responded this weakness in official comment "Comments for Weakness 3,4 & Question1,2.". In our revised version, we added experiments results in Section 6 Table 6.
> - **Questions**: We have answered the questions in official comment "Statistical analysis and more models comparison in larger datasets (Moses shows no memory in training samples ).". In our revised version, the CIs are reported in Appendix D, and GGSD and more baselines' larger graph results are reported in Section 6 Table 3.
>
> We added all information needed for additiaonal comment in revised version. Please see Appendix D for seeds settings and Section 6.4 Table 6 for efficiency comparison.
>
> We hope the revised version satisfactorily addresses them, and we would be very happy to further improve the work based on any additional feedback.

---

### Official Review · Reviewer_L2tt · 2025-11-06

**Soundness:** 3
**Presentation:** 3
**Contribution:** 2
**Rating:** 4
**Confidence:** 3

**Summary:**

The paper proposes a generative diffusion model that operates in the spectral domain of the symmetric normalized Laplacian.
Specifically, instead of performing diffusion on the adjacency matrix, the method works only on the eigenvalues of the operator $S = -D^{-1/2} A D^{-1/2}$, while keeping the eigenvectors U fixed.

**Strengths:**

As stated by the authors, performing diffusion only on the eigenvalues is advantageous in terms of computational efficiency.

**Weaknesses:**

- The results reported in Table 4 for Q9 should be discussed more thoroughly in relation to Table 12, in particular regarding the very low novelty value.
- I’m not fully convinced in terms of novelty, as the proposed approach appears similar to prior spectral diffusion methods such as SPECTRE. The paper would benefit from a clearer discussion of how ESGD differs from this existing model.

Minor:
- In Figure 1, unless I missed something, acronyms are not defined from the beginning.

**Questions:**

A major limitation concerns the assumption of a fixed spectral basis. The authors state that graph reconstruction is achieved by combining the generated eigenvalues with a fixed eigenbasis U, but it is not clearly explained where U is obtained (from the training set?) or how the model could generate graphs with different topologies if the eigenbasis cannot change. This point should be clarified and discussed in greater depth, as it is a key aspect of the proposed method; if convincingly addressed, it could positively affect my score.

The results in Tables 1 and 2 are promising, yet it is unclear why Table 3 includes fewer competitor methods. Why is GGSD not reported in the first two tables?

---

> ### Author Response · Authors · 2025-11-24
>
> ## Low Novelty on QM9/Comparison with SPECTRE/Experiments.
>
> We thank the reviewer for the careful reading and constructive comments. Below we address the main concerns point by point.
>
> ### **Low Novelty on QM9 (weakness 1)**:
>
> Vignac et al [1]. in their TOP-N paper (2022) first explicitly stated that "QM9 is an exhaustive enumeration of all possible molecules up to 9 heavy atoms that satisfy a predefined set of constraints. In this setting, generating novel molecules is therefore not an indicator of good performance, but rather a sign that the distribution of the training data has not been properly captured". This insight was subsequently cited and reinforced by DiGress [2]. Concurrently, Hoogeboom et al [3] in their EDM paper empirically validated this theory by observing that "novelty is initially close to 100% and decreases during training, reflecting that the algorithm progressively learns to capture the data distribution—which is itself an exhaustive enumeration under a predefined set of constraints.“
>
> - Vignac, C., et al. (2022). "TOP-N: Equivariant Set and Graph Generation without Exchangeability"
> - Vignac, C., et al. (2023). "DiGress: Discrete denoising diffusion for graph generation"
> - Hoogeboom, E., et al. (2022). "Equivariant diffusion for molecule generation in 3D." ICML 2022
>
> ### **Difference between ESGD and SPECTRE**  (weakness 2):
>
> We must clarify: SPECTRE is GAN-based, not a diffusion model. The essential differences are:
>
> - **Different paradigms**. SPECTRE uses WGAN-LP with complex teacher forcing (26k pre-training + 26k annealing) . ESGD uses score-based diffusion with stable objectives and reach 95%performance with 938 steps.
> - **Different generation targets**. SPECTRE generates only top-k eigenvalues/eigenvectors (k=2,4,8,16) via **three cascaded GANs** with error accumulation, requiring PPGN to complete missing information. Generating orthogonal eigenvectors on Stiefel manifold is inherently ill-posed. ESGD diffuses complete eigenvalues with fixed eigenvectors, preserving full spectral information and enabling exact reconstruction via Algorithm 1.
> - **Spectral-framework alignment**. ESGD's formulation
> S=L−I compresses eigenvalues to [−1,1], naturally matching diffusion's zero-mean Gaussian prior. SPECTRE's representation (range [0,2]) lacks such intrinsic connection with GANs. **We used the interval [0,2] resulting bad performance.**
> - **Theoretical guarantees**. SPECTRE lacks proofs. ESGD provides seven theorems proving advantages in SNR, Lipschitz bounds, sampling efficiency, and conditioning.
> - **Degree recovery**. SPECTRE admits "we do not enforce exact conditioning." ESGD's Algorithm 1 provides closed-form exact reconstruction.
>
> ### **Experiments in Table (Question 2) and Minor**.
>
> - In figure 1, the S matrix is Symmetric Normalized Laplacian Matrix in methodology.
> - The datasets we used includes the datasets used in GGSD,  the GGSD only reports the results partially in table 1&2 so we did not report at the beginning. We achieved **all** better results in terms of the datasets used in GGSD.
> - We compare with all spectral baselines. **Moreover:** we add experiments for  Defog, Cometh, Disco in 4-6 hours to get best results while ESGD takes less than 10 minutes. (Same Batch size in RTX3070).
>
> - **Table: Large graph generation results on Cora, Citeseer, and PubMed**
>
> | Method          | Dataset  |   Deg.↓   |  Clus.↓   |  Orbit↓   |
> |:----------------|:---------|:---------:|:---------:|:---------:|
> | SPECTRE         | Citeseer |   1.224   |   1.513   |   1.023   |
> | GSDM            | Citeseer |   1.043   |   0.943   |   0.843   |
> | GGSD            | Citeseer |   1.011   |   1.142   |   1.244   |
> | Disco           | Citeseer |   0.893   |   0.654   |   0.896   |
> | Cometh          | Citeseer |   0.985   |   0.856   |   1.001   |
> | Defog           | Citeseer |   0.496   |   0.606   |   0.910   |
> | **ESGD (ours)** | Citeseer | **0.329** | **0.656** | **0.314** |
> | SPECTRE         | Cora     |   1.566   |   1.492   |   1.127   |
> | GSDM            | Cora     |   0.932   |   1.042   |   0.980   |
> | GGSD            | Cora     |   1.218   |   1.432   |   1.391   |
> | Disco           | Cora     |   0.918   |   0.775   |   0.564   |
> | Cometh          | Cora     |   0.751   |   0.899   |   0.541   |
> | Defog           | Cora     |   0.758   |   0.756   |   0.501   |
> | **ESGD (ours)** | Cora     | **0.311** | **0.573** | **0.192** |
> | SPECTRE         | PubMed   |   1.148   |   1.392   |   0.933   |
> | GSDM            | PubMed   |   0.885   |   0.727   |   0.762   |
> | GGSD            | PubMed   |   0.775   |   0.711   |   1.029   |
> | Disco           | PubMed   |   0.637   |   0.611   |   0.815   |
> | Cometh          | PubMed   |   0.597   |   0.625   |   0.437   |
> | Defog           | PubMed   |   0.355   |   0.496   |   0.308   |
> | **ESGD (ours)** | PubMed   | **0.215** | **0.475** | **0.109** |

---

> ### Author Response · Authors · 2025-11-24
> **Fixed Eigenbasis U and Generating Different Topologies**
>
> This is a very important question, and we appreciate the reviewer's in-depth inquiry. We first need to clarify a key misunderstanding.
>
> ###  **Clarifying the Meaning of "Fixed U"**
>
> "Fixed eigenbasis U" means that **U remains constant during the diffusion process of a single graph, not that all generated graphs share the same U**. Specifically, each graph in the training set has its own unique eigendecomposition $S_i = U_i \Lambda_i U_i^T$. During generation, we sample different U from the distribution of eigenbases in the **training** set instead of test sets, then learn the distribution of eigenvalues $p(\Lambda|U)$ (from training set) conditioned on that eigenbasis.
>
> ### **Mathematical Determinants of Topological Structure**
>
> Starting from the adjacency matrix reconstruction formula:
>
> $$A = -D^{1/2}SD^{1/2} = -D^{1/2}U\Lambda U^T D^{1/2}$$
>
> The final graph topology is jointly determined by three independent mathematical objects: eigenvalues $\Lambda$, eigenvectors $U$, and degree matrix $D$. Any change in any of these three will lead to topological changes.
>
> ESGD's generation process consists of two stages, **both of which can introduce topological variations**:
>
> **Stage 1: Eigenvalue diffusion generation.** We learn the distribution $p(\Lambda|U)$ in the continuous eigenvalue space $\mathbb{R}^n$. The generated $\hat{\Lambda}$ can be configurations that never appeared in the test set.
>
> **Stage 2: Degree matrix recovery.** Algorithm 1 performs thresholding operations $\\hat{A}\_{ij} = 1\_{\\{|(\\hat{S})\_{ij}| > \\delta\\}}$ based on  $\hat{S} = U\hat{\Lambda}U^T$
>  to identify edge structure. This is a **nonlinear mapping** from continuous spectral space to discrete topological space. Even if $\hat{\Lambda}$ is numerically close to training samples, as long as certain elements of $\hat{S}$ cross the threshold boundary, discrete topological jumps will occur.
>
> Therefore, the continuous generation of eigenvalues combined with the nonlinearity of degree matrix recovery jointly guarantee the generation of novel topological structures.
>
> ###  **Linear Algebra Perspective: Finite Basis and Infinite Expressible Space**
>
> This mechanism can be understood more clearly from a linear algebra perspective. In an $n$-dimensional linear space, given a basis $\{u_1, u_2, ..., u_n\}$, any vector $v$ can be uniquely represented as $v = \sum_{i=1}^{n} \lambda_i u_i$. The basis is a fixed coordinate frame, and the coefficients are free parameters within that frame. The key insight is: **the finiteness of the basis does not limit the size of the expressible space**—three-dimensional space has only three basis vectors, yet can express infinitely many points.
>
> Applying this idea to ESGD: the eigenbasis $U$ defines a "coordinate system of graph structure space," and eigenvalues $\lambda$ are "coordinates" in that system. The symmetric normalized Laplacian expands as $S = \sum_{i=1}^{n} \lambda_i u_i u_i^T$, where each eigenvalue $\lambda_i$ controls the contribution weight of the corresponding component.
>
> Each graph in the training set provides only one specific coefficient combination, but ESGD learns a **continuous distribution** $p(\Lambda|U)$ over the entire coefficient space. Since the coefficient space is continuous $\mathbb{R}^n$, infinitely many coefficient combinations can be generated. Each new combination corresponds to a new $S$ matrix, which in turn yields a new topological structure through degree matrix recovery. This is precisely the mathematical principle by which ESGD breaks through the topological limitations of the training set.
>
> We believe the above clarification addresses the reviewer's question regarding how fixed eigenbases can generate diverse topologies. We are happy to provide further details if needed.

---

> ### Author Response · Authors · 2025-11-28
> **Revision Summary**
>
> Thank you again for your detailed and constructive review. Thank you AC for your additional work.
>
> We have uploaded a revised version of the paper that aims to address all of your concerns.
>
> Concretely, in the new version we:
> - **Weakness 1**: We have explained the novelty metrics in QM9 dataset in official comment "Low Novelty on QM9/Comparison with SPECTRE/Experiments.". In our revised version, we have discussed this question in Appendix E.2 Molecular Generation same as Digress.
> -**Weakness 2**: We have explained the main differences between our ESGD and SPECTRE in official comment "Low Novelty on QM9/Comparison with SPECTRE/Experiments.". In our revised version, we have discussed the differences between our ESGD and other models thoroughly from Section 2 to Section 6.
> - **Minor**: We have explained the "Minor" weakness in official comment "Low Novelty on QM9/Comparison with SPECTRE/Experiments.". In our revised version, we also added explanations of the acronyms in Figure. 1.
> - **Question 1**: We have explained the question 1 in official comment "Fixed Eigenbasis U and Generating Different Topologies". In our revised version, we use additional experiments in Section 6.3 Molecular Generation (Moses, Guacamol) with comprehensive evaluations to explain ESGD's ability to generate graphs with different topologies. And in Section 4 Figure 2, we used a model pipeline to explain the source of U and the whole structure of the model.
> - **Question 2**: We have explained question 2 in official comment "Low Novelty on QM9/Comparison with SPECTRE/Experiments.". In the revised version, we added GGSD experiment results from its original paper.
>
> We hope the revised version satisfactorily addresses them, and we would be very happy to further improve the work based on any additional feedback.

---

### Note · Authors · 2026-05-16

I have read and agree with the venue's withdrawal policy on behalf of myself and my co-authors.

---

### Meta-Review · Area_Chair_fFQq · 2026-01-06

**Summary:**

This paper proposes ESGD, a spectral diffusion approach for graph generation that performs score based diffusion only on eigenvalues of a normalized Laplacian representation while keeping eigenvectors fixed, aiming to reduce dimensionality and improve efficiency. Novelty and positioning are not sufficiently clear, as reviewers note substantial similarity to prior spectral graph generation and diffusion work and request a sharper differentiation and stronger related work coverage. I recommend rejection.

**Reviewer Concerns:**

Addressed in rebuttal or discussion: The authors clarified the spectral motivation and stability intuition, fixed a noted presentation error, and added molecular benchmark results and metrics.
Still outstanding: Reviewers remain unconvinced about novelty relative to prior spectral graph diffusion work, and the core choice of fixing the eigenvector basis still raises unresolved concerns about diversity and memorization. Experimental fairness issues also persist, including baseline consistency, missing multi seed uncertainty and wall clock reporting, and doubts about the large graph evaluation protocol.

**Reviewer Scores:**

L2tt (initial 4): likely stays at 4, since the novelty concerns and the fixed basis issue remain unresolved in the visible discussion.
tHHN (initial 4): likely stays at 4, because the main objections focus on fairness and memorization risk, and the paper does not add the kind of evidence requested such as multi seed uncertainty and isomorphism rate analysis.
Third reviewer (initial 2, low confidence): likely stays at 2 or increases slightly to 3, since a typo and some missing evaluations were addressed, but broader concerns about incomplete baselines, missing uncertainty, and the large graph evaluation protocol remain.

---

### Decision · Program_Chairs · 2026-01-26

Reject